# A distributed fMRI-based signature for the subjective experience of fear

Feng Zhou[1,2,3], Weihua Zhao[1], Ziyu Qi[1], Yayuan Geng[1], Shuxia Yao[1], Keith M. Kendrick [1], Tor D. Wager [2,4✉] & Benjamin Becker [1,4✉]

The specific neural systems underlying the subjective feeling of fear are debated in affective neuroscience. Here, we combine functional MRI with machine learning to identify and evaluate a sensitive and generalizable neural signature predictive of the momentary self-reported subjective fear experience across discovery (n = 67), validation (n = 20) and generalization (n = 31) cohorts. We systematically demonstrate that accurate fear prediction crucially requires distributed brain systems, with important contributions from cortical (e.g., prefrontal, midcingulate and insular cortices) and subcortical (e.g., thalamus, periaqueductal gray, basal forebrain and amygdala) regions. We further demonstrate that the neural representation of subjective fear is distinguishable from the representation of conditioned threat and general negative affect. Overall, our findings suggest that subjective fear, which exhibits distinct neural representation with some other aversive states, is encoded in distributed systems rather than isolated 'fear centers'.

---

[1] Clinical Hospital of Chengdu Brain Science Institute, MOE Key Laboratory for Neuroinformation, University of Electronic Science and Technology of China, Chengdu, China. [2] Department of Psychological and Brain Sciences, Dartmouth College, Hanover, NH, USA. [3] Faculty of Psychology, Southwest University, Chongqing, China. [4] These authors contributed equally: Tor D. Wager, Benjamin Becker. ✉email: Tor.D.Wager@Dartmouth.edu; ben_becker@gmx.de

Fear is probably the most studied emotion during the last decades, yet despite considerable advances in animal models and human neuroimaging research, vigorous debates on how to define and investigate fear and its facets continue[1–4]. When we talk about fear in everyday life, we primarily refer to the subjective feeling of being afraid[3]. However, in psychological and neuroscientific conceptualizations, fear also describes defensive behaviors, such as freezing, and peripheral physiological changes that accompany such behaviors[5,6].

The neural basis of 'fear', or threat behaviors, has been extensively mapped in animal models using Pavlovian conditioning and predator exposure protocols[3,7]. These models provide compelling evidence for a pivotal role of subcortical systems, particularly the central extended amygdala, as well as the hypothalamus and periaqueductal gray (PAG), in mediating threat detection and defensive responses[7–13]. However, the subjective emotional experience of fear remains ultimately inaccessible in animal models, and recent conceptual frameworks argue that the evolutionarily conserved defensive survival circuits that account for the behavioral and physiological responses to threats might be distinct from those underlying the subjective experience of fear[3,6,14–16]. The differentiation between the defensive response and the subjective experience of fear has critical implications for translational research on pathological fear[6], given that animal models primarily evaluate novel treatments by means of effects on physiological and behavioral defensive threat reactivity[17], whereas feelings of exaggerated fear or anxiety represent the primary clinical outcome and reason for patients to seek treatment[18].

In humans, lesion and functional magnetic resonance imaging (fMRI) approaches have been employed to determine the specific brain systems that underlie the subjective feeling of fear. Early studies in a patient with focal amygdala lesions demonstrated impairments in fear-related processes, including recognition and experience of fear[19,20], which contributed to an amygdala-centric fear perspective. However, subsequent studies reported variable fear-related functional consequences in patients with focal amygdala lesions[21]. For instance, some patients with focal and complete amygdala lesions maintain intact fear recognition[22] and experience fear, anxiety and panic in response to breathing $CO_2$-enriched air[23]. fMRI studies in healthy subjects suggest that it is time to move beyond an amygdala-centric fear perspective and demonstrate that stimuli that evoke subjective feelings of fear elicit activation not only in the amygdala but also PAG, hypothalamic and frontal regions[24–27]. However, the conventional fMRI approach applied in these studies has been limited. In particular, it is designed to permit the inference of whether a single brain region (or voxel) is activated conditionally on a stimulus, but does not allow reverse inferences about 'fear' states given brain activity[28]. Furthermore, mass univariate approaches are inherently focused on individual regions or, in the case of connectivity analyses, circumscribed networks yet do not model joint activity across distributed brain regions working together to underpin fear experience[29], and stimulus-induced activation changes in single brain regions typically have only modest effect sizes[30–32]. These issues raise the question of whether isolated regions can provide adequate and comprehensive brain-level descriptions of complex mental processes such as the subjective feeling of fear.

To provide sufficient and process-specific brain-level descriptions of mental processes with large effect sizes recent studies have combined fMRI with machine-learning-based multivariate pattern analysis (MVPA). This approach can capture information at much finer spatial scales[33] and provide considerably larger effect sizes in brain-outcome associations[31] thus allowing the development of sensitive and specific brain signatures of mental processes[32,34–36], including signatures for acquired defensive responses[37] and subjective emotional states[38,39]. Moreover, an initial MVPA study has revealed promising findings suggesting that offline categorical fear ratings collected before fMRI are associated with a neural signature that is independent of online autonomic arousal indices acquired during fMRI[40] (henceforward referred to as animal fear schema signature (AFSS) in the present study for convenience). The MVPA approach additionally allows functional separation of mental processes based on population coding[31], despite overlapping univariate activation[35,41,42] and thus offers an opportunity to determine process-specific neural representations of (often) concurrent fear-related processes, such as the experience of fear and defensive responses.

Moreover, the perspective of an isolated fear center in the brain has additionally been challenged by conceptual perspectives, including recent appraisal[43] and constructionist[2] theories of emotion which suggest that emotional experiences result from interactions between multiple systems including core affect, sensory, memory, motor, and cognitive systems[44], and by the two-system model suggesting that interactions of subcortical defensive systems with prefrontal regions engaged in consciousness are critical to establish a neural representation of the subjective fear experience[6].

Here, in the context of ongoing debates about the neural representations of fear, we capitalize on recent advances in MVPA-based neural decoding techniques to determine whether (1) it is possible to develop a sensitive and generalizable neural representation of the subjective fear experience on the population level, (2) this neural representation can predict momentary (trial-wise) fear experience on the individual level, (3) the neural representation in isolated systems such as the amygdala or 'cortical consciousness network' is sufficient to capture the subjective experience of fear, and (4) the neural representation of the momentary fear experience is distinct from the representations of the conditioned defensive threat response and general aversive states. More specifically, we employed a support vector regression (SVR) algorithm in healthy participants ($n = 67$) to identify the brain signature that predicted the intensity of trial-by-trial rated subjective experiences of fear elicited by fear-evoking pictures ranging from low to high fear induction (Fig. 1a). The performance of the established visually induced fear signature (VIFS) was evaluated in (a) an independent validation cohort ($n = 20$), who underwent a similar but not identical fear induction paradigm (Fig. 1b) as well as a generalization cohort ($n = 31$) from a previous study that employed a different fear induction paradigm and MRI system[40] and (b) a comparison with the AFSS (Fig. 1c). To extend the perspective from a population to an individual level we tested whether the VIFS can predict trial-wise fear experience for each subject in discovery and validation cohorts separately. We further systematically identified brain regions that were associated with (forward model, i.e., expressing the observed data as functions of underlying variables) and predictive of (backward model, i.e., expressing variables of interest as functions of the data) subjective fear experience[45] and examined to what extent single brain systems or networks can capture subjective fear experience. Moreover, to determine the functional specificity of the neural fear experience signature we compared the spatial and functional similarities between the VIFS with the signature of conditioned defensive threat response[37] (mostly referred to as 'conditioned fear response' in the literature, but see ref.[3] for a discussion on the terminology) and general negative emotional experience[32], respectively (Fig. 1c). Together this systematic evaluation can advance ongoing debates on how the brain constructs subjective fear, whether the neural mechanisms of the conscious experience of fear are distinct from defensive responses

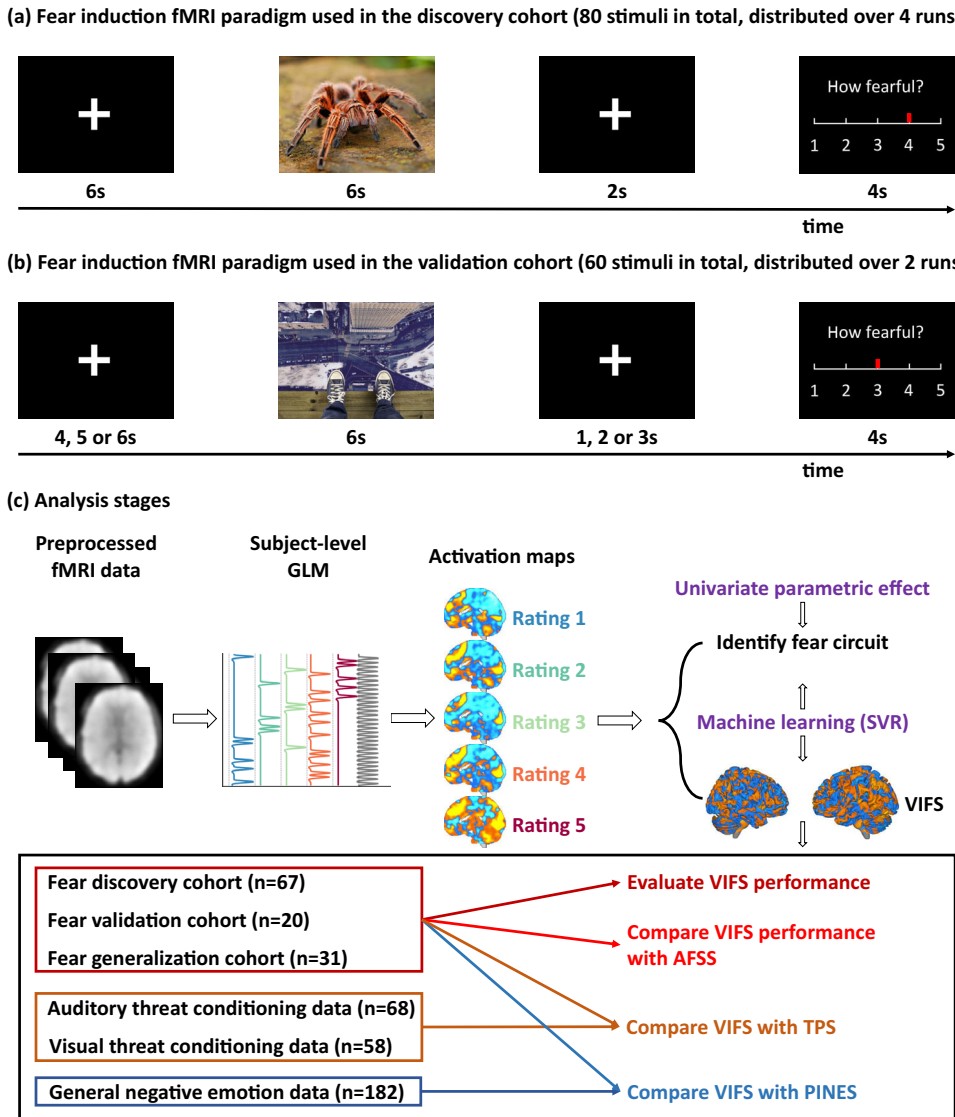

**(a) Fear induction fMRI paradigm used in the discovery cohort (80 stimuli in total, distributed over 4 runs)**

**(b) Fear induction fMRI paradigm used in the validation cohort (60 stimuli in total, distributed over 2 runs)**

**(c) Analysis stages**

**Fig. 1 Experimental paradigms and analysis stages.** Discovery (**a**) and validation (**b**) cohorts underwent two similar but not identical fear induction and rating paradigms during fMRI acquisition. Of note, examples of the fear-evoking photos are pictures only for display purposes and not included in the original stimulus set. The pictures have been obtained from pixabay.com under the Pixabay License, and are free for commercial and noncommercial use across print and digital. **c** depicts the analytic stages and datasets used in the present study. Specifically, a whole-brain multivariate pattern predictive of the level of subjective experience of fear was trained on the discovery sample ($n = 67$) using support vector regression and further evaluated in discovery (cross-validated), validation ($n = 20$), and generalization ($n = 31$) cohorts. We next systematically applied univariate and multivariate analyses to determine the spatial scale and local contributions of specific regions to the momentary subjective fear representation. Finally, we tested whether subjective fear was encoded with a neural signature that was distinct from the representation of conditioned threat (CS+ versus CS− cues) and general negative affect. GLM general linear model, SVR support vector regression, VIFS visually induced fear signature developed in the current study, AFSS animal fear schema signature developed by Vincent Taschereau-Dumouchel and colleagues, TPS threat-predictive signature developed by Reddan and colleagues, PINES picture-induced negative emotion signature developed by Luke Chang and colleagues. See 'Methods' and 'Results' for the details of the datasets and brain signatures used in this study.

elicited by conditioning[3,6,14–16] or unspecific aversive emotional experience.

In this work we develop a sensitive and generalizable neural signature predictive of the momentary subjective fear experience and systematically demonstrate that accurate fear prediction crucially requires distributed brain systems, with important contributions from cortical (e.g., prefrontal, midcingulate and insular cortices) and subcortical (e.g., thalamus, periaqueductal gray, basal forebrain, and amygdala) regions. The neural representation of subjective fear are distinguishable from the neural representations of conditioned threat and general negative affect. Overall, the findings suggest that subjective fear experience exhibits a distinguishable neural representation from some other aversive states and is encoded in distributed systems rather than isolated 'fear centers'.

## Results

**Visual stimuli elicited a robust range of subjective fear**. The experience of fear was induced by visual stimuli with varying levels of subjective fear induction e.g., by stimuli showing threatening or dangerous situations. Subjects were explicitly instructed to imagine that they were encountering the situation displayed in the picture to increase the vividness of the stimulus and were asked to report their current level of fear for each trial

on a 5-point Likert scale ranging from 1 (neutral/slightest fear) to 5 (very strong fear). To initially test whether the visual stimuli elicited meaningful and varying levels of subjective fear, we plotted the number of each selected subjective fear level (across subjects) for each run (Supplementary Fig. 1a) and for each stimulus category (animal, human and scene; Supplementary Fig. 1b). We found that the stimuli induced sufficient levels of fear experience in the discovery cohort ($n = 67$) which was used to develop the neural signature of subjective fear (see below for details), such that over 14% trials of each stimulus type were rated as 5 (reflecting that they induced strong fear) and self-reported fear levels were generally evenly distributed across categories and runs. Moreover, 65 out of 67 subjects reported all 5 levels of subjective fear whereas the remaining 2 subjects used ratings '1–4'.

**A brain signature sensitive to predict visually induced subjective experience of fear.** We applied SVR to identify a whole-brain signature of fMRI activation that predicted the intensity of self-reported fear ratings during observation of fear-evoking pictures in the discovery cohort (Fig. 2a). To evaluate the performance of the visually induced fear signature (VIFS), we applied the VIFS to data from test subjects in both discovery (10 × 10-fold cross-validated, see 'Methods' for details) and validation ($n = 20$) cohorts to calculate the VIFS pattern expressions for individual participants' activation maps for each of five levels of reported fear. The developed VIFS accurately predicted ratings

of reported fear in both discovery and independent validation cohorts. Specifically, for individual participants in the discovery cohort the average within-subject correlation between predicted and actual fear ratings (5 or 4 pairs of scalar values per subject) was $r = 0.89 \pm 0.01$ (standard error (SE)), the mean explained variance score (EVS) was $72.5 \pm 2.1\%$, the average root mean squared error (RMSE) was $1.38 \pm 0.08$ and the overall (between- and within-subjects) prediction-outcome (i.e., 333 pairs) correlation coefficient was 0.57 (averaged across ten repetitions; EVS = 17%; bootstrapped 95% confidence interval (CI) = [0.49, 0.63]) (Fig. 2b). Testing the VIFS model developed on the discovery cohort, with no further model fitting, in the validation cohort (Fig. 1b) revealed comparably high prediction−outcome correlations (within-subject $r = 0.87 \pm 0.02$; mean EVS = $68.3 \pm 5.6\%$; average RMSE = $1.40 \pm 0.14$; overall prediction−outcome $r = 0.59$, 95% CI = [0.48, 0.69], EVS = 12%, permutation test one-tailed $P < 0.001$; Fig. 2c), indicating a sensitive and robust subjective fear signature on the neural level (see also generalization and benchmarking of the VIFS below). To further determine the sensitivity of the VIFS to predict levels of subjective fear experience a two-alternative forced-choice test was applied, comparing pairs of activation maps within each subject and choosing the one with higher VIFS response as more fearful. The VIFS response accurately classified high (average of rating 4 and 5) versus moderate (rating 3) and moderate versus low (average of rating 1 and 2) fear in both cohorts with 88−93% accuracy (Cohen's $d$: 1.18–1.40), and high versus low with 100% accuracy in both cohorts (Cohen's $d$: 2.20–2.58) (Fig. 1b, c; see also Table 1

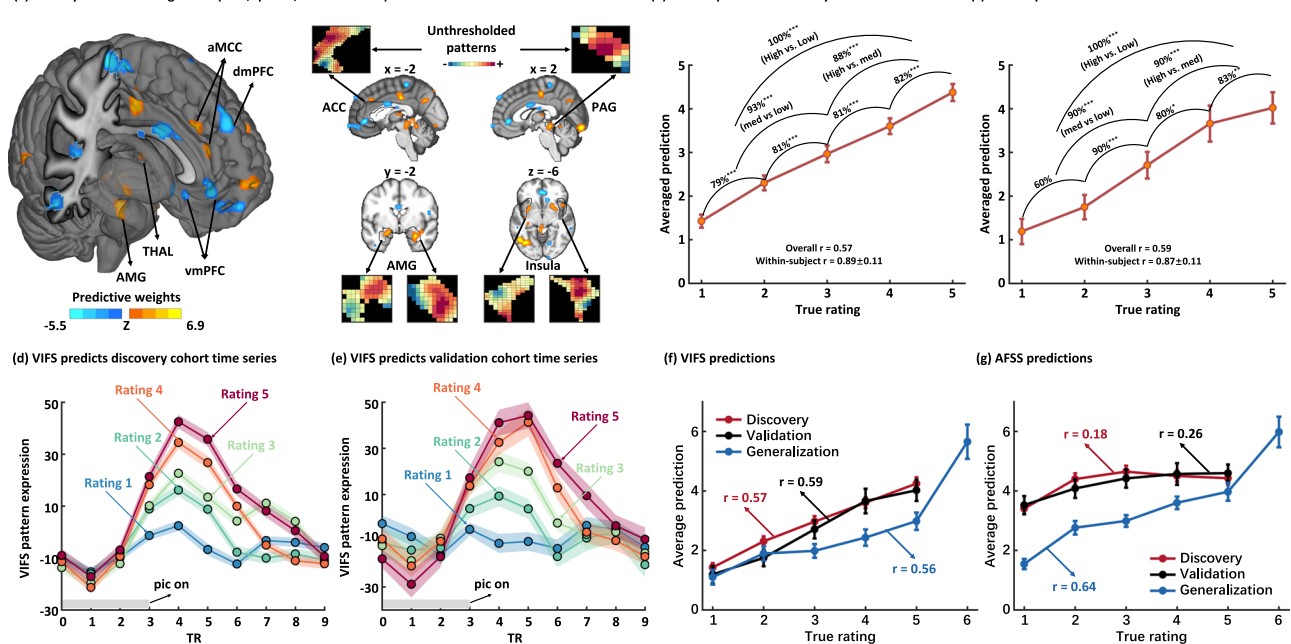

**Fig. 2 Visually induced fear signature (VIFS). a** depicts the VIFS pattern thresholded using a 10,000-sample bootstrap procedure at $q < 0.05$, FDR corrected. Inserts show the spatial topography of the unthresholded patterns in the left ACC, right PAG, bilateral AMG, and bilateral insula. AMG denotes amygdala, THAL thalamus, vmPFC ventromedial prefrontal cortex, dmPFC dorsal medial prefrontal cortex, dACC dorsal anterior cingulate cortex, MCC middle cingulate cortex, ACC anterior cingulate cortex, and PAG periaqueductal gray. **b, c** depict the predicted fear experience (subjective ratings; mean ± SE) compared to the actual level of fear for the cross-validated discovery cohort and the independent validation cohort, respectively. Accuracies reflect forced-choice comparisons. Two-sided binomial tests were used to test whether the classification accuracies were higher than chance level. $r$ indicates the Pearson correlation coefficient between predicted and true ratings. **d, e** depict an average peristimulus plot (mean ± SE) of the VIFS response to the cross-validated discovery cohort and the independent validation cohort. This reflects the average VIFS response at every repetition time (TR; 2 s) in the time series separated by the fear ratings. Of note, the VIFS reacts with a latency of approximate 4 s after stimulus onset which corresponds to the timing of the hemodynamic response function (HRF) following stimulus onset. **f, g** compare the fear prediction (mean ± SE) of AFSS with the VIFS on discover, validation, and generalization cohorts, respectively. * indicates $P < 0.05$, **$P < 0.01$ and ***$P < 0.001$ (uncorrected). Error bars and shaded regions indicate SEs. VIFS visually induced fear signature, AFSS animal fear schema signature, SE standard error of mean. Source data are provided as a Source Data file.

for a detailed summary of classification performance). Moreover, the VIFS response could distinguish each successive pair of fear rating levels (e.g., rating 4 versus 5) with ≥80% accuracy, which were significantly better than chance level (50%; $P < 0.001$; except ratings of '1' versus '2' in the validation cohort) (Fig. 2b, c).

Retraining the decoder excluding the occipital lobe revealed high prediction accuracies, suggesting that although the fear-predictive signals might be partly embedded in regions engaged in visual processing the contribution of visual cortical patterns is small (Supplementary Results and Supplementary Fig. 2; see also the prediction using visual network alone in the following 'Alternative models to determine the contribution of isolated fear predictive systems' section, which demonstrated a substantial lower performance as compared to the whole-brain prediction). In addition, we applied the VIFS to time series data using dot-product in the discovery ($10 \times 10$-fold cross-validated) and validation datasets to determine the specificity of the visually induced fear pattern with respect to confrontation with imminent threat (rather than anticipation or cognitive evaluation). Visual inspection of the VIFS reactivity at each timepoint following stimulus onset indicated that the VIFS response began approximately 4 s following picture onset and increased with increasing levels of reported fear during approximately 6–12 s (Fig. 2d, e). These findings validate the adequacy of the hemodynamic response model and confirmed that the VIFS was specific to brain activity during threat exposure, as opposed to threat anticipation (pre-stimulus) or cognitive evaluation (response reporting).

**Generalization and benchmarking of VIFS performance.** An important feature of population-level neural signatures is that their performance can be evaluated in new datasets and paradigms, although prediction across cohorts, paradigms and different MRI systems has been challenging. Taschereau-Dumouchel et al.[40] developed a neural decoder which predicted the general subjective fear of different animal categories (assessed before fMRI) and the authors shared the dataset used for training their model—which we term the 'generalization dataset' here—allowing us to compare the performance of the VIFS with the AFSS on the discovery, validation and generalization cohorts. We found that the VIFS predicted all three datasets well (overall prediction−outcome correlations $rs > 0.56$) while the AFSS only performed well on its training dataset (overall prediction−outcome correlation $r = 0.64$) but poorly on both discovery and validation cohorts (overall prediction−outcome correlation $rs < 0.27$) (Fig. 2f, g; Table 1; see also Supplementary results for details), indicating a robust generalization and high sensitivity of the VIFS to predict fear experience across populations, fear induction paradigms and MRI systems.

**Within-subject trial-wise prediction.** The feeling of fear is a momentary, highly subjective and individually constructed state[2,4] and thus a key question is to what extent the population-level model (i.e., the VIFS), which is a statistical summary of a highly variable set of instances, can predict momentary (trial-wise) fear experience for each subject (on the individual level). To this end we performed single-trial analyses using the Least Squares All (LSA) approach[46] to obtain a beta map for each stimulus for each subject in both discovery (~80 beta maps per subject) and validation (~60 beta maps per subject) cohorts (see 'Methods' for details). The VIFS was next applied to these beta maps to calculate the pattern expressions which were further correlated with the true ratings for each subject separately. The statistical significance was evaluated by prediction−outcome Pearson correlation for each subject separately. We found that the VIFS could significantly predict trial-by-trial ratings for 61 out of 67 subjects in the discovery cohort (cross-validated) and for 16 out of 20 subjects in the validation cohort. The mean prediction−outcome correlations were $0.38 \pm 0.01$ and $0.40 \pm 0.03$ for the discovery and validation cohorts, respectively. Our findings suggest that although fear experience differs between individuals[2,4] the VIFS could predict the level of momentary fear experience on the individual level in a large population.

**Subjective fear is associated with and predicted by distributed neural systems.** We systematically determined individual brain regions that were associated with subjective fear ratings and that provided consistent and reliable contributions to the whole-brain fear decoding model using different analytic strategies. We first examined the conventional univariate linear parametric effect of fear ratings, i.e., voxels that increased (yellow in Supplementary Fig. 3a) or decreased (blue in Fig. Supplementary 3a) linearly with within-subject fear ratings across trials, by performing one-sample $t$ test on the parametric modulation beta maps. Subjective fear was associated with activation in a broad set of cortical and subcortical regions, including increased activation in the amygdala and surrounding sublenticular extended amygdala, anterior insula, anterior midcingulate cortex (aMCC), thalamus, PAG and surrounding midbrain, ventrolateral prefrontal and lateral orbitofrontal cortices, and fusiform/ventral occipital-temporal regions. Conversely, we found negative correlations with fear ratings in the ventromedial prefrontal cortex (vmPFC), medial orbitofrontal cortex (OFC), posterior insula/operculum, and dorsolateral prefrontal cortex (dlPFC), posterior cingulate cortex (PCC), inferior parietal lobule (IPL) and supplementary motor area (SMA) ($q < 0.05$, FDR corrected; Supplementary Fig. 3a).

We then compared these univariate (single-voxel) findings to multivariate models in several ways. First, we performed a one-sample $t$ test analysis (treating participant as a random effect) on

---

**Table 1 Comparing performance of VIFS with AFSS.**

| Classifications | | VIFS | AFSS |
|---|---|---|---|
| Discovery dataset | high versus low | $100 \pm 0\%^{***}$ (2.58) | $79 \pm 5.0\%^{***}$ (0.77) |
| | high versus moderate | $88 \pm 3.9\%^{***}$ (1.18) | $42 \pm 6.0\%^{NS}$ (−0.22) |
| | moderate versus low | $93 \pm 3.1\%^{***}$ (1.40) | $84 \pm 4.5\%^{***}$ (1.02) |
| Validation dataset | high versus low | $100 \pm 0\%^{***}$ (2.20) | $95 \pm 4.9\%^{***}$ (1.88) |
| | high versus moderate | $90 \pm 6.7\%^{***}$ (1.21) | $55 \pm 11.1\%^{NS}$ (0.19) |
| | moderate versus low | $90 \pm 6.7\%^{***}$ (1.27) | $65 \pm 10.7\%^{NS}$ (0.78) |
| Generalization dataset | high versus low | $87 \pm 6.2\%^{***}$ (1.10) | $90 \pm 5.5\%^{***}$ (1.56) |
| | high versus moderate | $83 \pm 6.8\%^{***}$ (0.97) | $87 \pm 6.2\%^{***}$ (0.79) |
| | moderate versus low | $83 \pm 7.0\%^{***}$ (0.86) | $93 \pm 4.7\%^{***}$ (1.52) |

For each dataset we used VIFS and AFSS to classify high, moderate, and low subjective fear using two-alternative forced-choice tests. Performance was shown as accuracy ± SE (Cohen's $d$). *** denotes uncorrected $P < 0.001$, and $^{NS}$ denotes non-significant based two-sided binomial tests. Source data are provided as a Source Data file.
VIFS visually induced fear signature, AFSS animal fear schema signature.

the weights from within-subject (ideographic) multivariate predictive models (for details, see 'Methods'). Like the univariate maps, within-subject predictive models (backward models) included consistent weights in brain regions spanning multiple large-scale cortical and subcortical systems, which exhibited a large overlap with the fear regions as determined by the univariate approach (Supplementary Fig. 3b; $q < 0.05$, FDR corrected).

Some brain features that make large contributions to the multivariate models might capture and control for sources of noise in the data, rather than being directly related to mental events[45]. To provide a more transparent comparison between univariate and multivariate results, we thus calculated within-subject reconstructed 'activation patterns' (forward models; see 'Methods' for details), which assess the relationships between each voxel and the response (fitted values) in the multivariate model. These are also referred to as 'structure coefficients' in the statistical literature[47]. Supplementary Fig. 3c shows results of a group analysis of 'activation patterns' across individuals ($q < 0.05$, FDR corrected). As Haufe et al.[45] suggest, voxels that exhibit significant predictive weights and structure coefficients are important regions that are both directly correlated with the outcomes (i.e., fear ratings) and are predictive after accounting for other brain regions in the multivariate model. As shown in Supplementary Fig. 3c, the thresholded 'model activation pattern' was remarkably similar to the univariate parametric effects of fear ratings (Supplementary Fig. 3a). This suggests that the multivariate model is encoding activity across distributed regions and confirms that subjective fear is associated with activity in a large number of cortical and subcortical regions. Indeed, a formal assessment of overlap (Supplementary Fig. 3d) showed that virtually all regions with consistent, significant model weights in the multivariate models also encoded model information (i.e., showed significant 'model activation patterns'). The broad conclusion is that the neural representation of human fear is not limited to a single or a set of focal regions (e.g., the amygdala), but rather includes a broad set of regions spanning multiple systems.

We next determined regions that reliably contributed to the fear prediction within the VIFS itself by applying a bootstrap test to identify regions with significant, consistent model weights ($q < 0.05$, FDR corrected) [34]. In line with within-subject models, a set of distributed brain regions exhibited significant model weights (Fig. 3a) and structure coefficients (Fig. 3b), including amygdala, MCC, insula, inferior frontal cortex (IFG), PAG, occipital and sensorimotor areas (Fig. 3c).

Overall, regions that were most consistently associated with subjective fear across the analyses included key regions engaged in conditioned threat (amygdala, aMCC and PAG), and general avoidance motivation (anterior insula, posterior ventral striatum) as determined across species while other regions such as the right posterior lateral prefrontal cortex/inferior frontal junction and ventral occipito-temporal stream have been associated with cognitive emotion construction in humans and dysregulated emotional experience in mental disorders[32,48]. Negative associations with fear were most consistently identified in medial prefrontal and sensorimotor regions. In conclusion, across both univariate and multivariate analyses, our results indicate that fear experience is represented in distributed neural systems involved in defensive responses, avoidance behavior, negative affect, emotional awareness as well as pathological fear and anxiety.

**Alternative models to determine the contribution of isolated fear-predictive systems: local searchlights, pre-defined regions and networks perform considerably worse than VIFS.** Given the continuous debate on the contribution of specific brain regions, such as amygdala[8,13,20,49] and more recently the cortical

consciousness network[3,6], to the subjective fear experience, we (1) located local brain regions that were predictive of subjective fear experience using both searchlight- and parcellation-based analyses, and (2) examined to what extent models trained on single brain region and network could predict subjective fear ratings as compared with the whole-brain model (i.e., the VIFS). As shown in Fig. 4a, b, subjective fear experience could be significantly predicted by activations in widely distributed regions (averaged across $10 \times 10$-fold cross-validation procedure). Given that the uncorrelated $P$ values equivalent to $q < 0.05$ were liberal in this case we displayed brain regions that survived at $P < 0.001$ uncorrected (corresponding to $q < 0.004$ and 0.003, FDR corrected, respectively). However, none of these local regions predicted subjective fear to the extent the VIFS did (see also Supplementary Fig. 4a−d for predictions of models trained on discovery cohort on validation and generalization cohort).

We next re-trained predictive SVR models (with the identical cross-validation and prediction procedure as used for the VIFS) restricted to activations in (a) the bilateral amygdala; (b) a pre-defined cortical network associated with consciousness[6] (see 'Methods' for details); (c) a subcortical region group (including striatum, thalamus, hippocampus and amygdala); and (d) each of seven large-scale resting-state functional networks[50,51]. We found that the amygdala (prediction−outcome correlation $r = 0.26$, 0.25, and 0.32 for discovery (cross-validation), validation, and generalization cohorts, respectively) as well as the other brain networks (see Fig. 4c, d, Supplementary Table 1 and Supplementary Fig. 4e, f see details) could, to some extent, predict subjective fear ratings. However, although statistically significant ($Ps < 0.001$, one-tailed permutation $t$ tests) the effect sizes in terms of prediction−outcome correlations (including searchlight- and parcellation-based predictions) were substantially smaller than those obtained from the VIFS, which used features that span multiple systems.

This comparison is fair even though the number of features differ, as models were always tested on holdout participants, eliminating the problem of overfitting as more predictors are used (a substantial problem in models trained and tested without cross-validation). However, to assess the potential effect of the numbers of features in the prediction analyses (i.e., whole-brain model uses much more features/voxels), as shown in Fig. 4d, we randomly selected samples from a uniform distribution spanning the entire brain (black), consciousness network (red), subcortical regions (light purple) or a single resting-state network (averaged over 1000 iterations)[31]. The asymptotic prediction when sampling from all brain systems as we did with the VIFS (black line in Fig. 4d and Supplementary Fig. 4e, f) was substantially higher than the asymptotic prediction within individual networks (colored lines in Fig. 4d and Supplementary Fig. 4e, f; see also Supplementary Table 1 for details). This analysis thus demonstrated that whole-brain models have much larger effect sizes than those using features from a single network. Furthermore, model performance reached asymptote when approximately 10,000 voxels were randomly sampled across the whole-brain, as long as voxels were drawn from multiple systems, further emphasizing that subjective fear experience is encoded in distributed neural patterns that span multiple systems. Importantly, we found similar results when applying the models trained on the discovery cohort to the validation and generalization cohorts, indicating that models trained on ~10,000 randomly sampled voxels were robust and generalizable. Together the results from the systematic analyses provide the first evidence that the subjective experience of fear is represented in distributed neural systems which argues against fear experience being reducible to activations in any single brain region or canonical network.

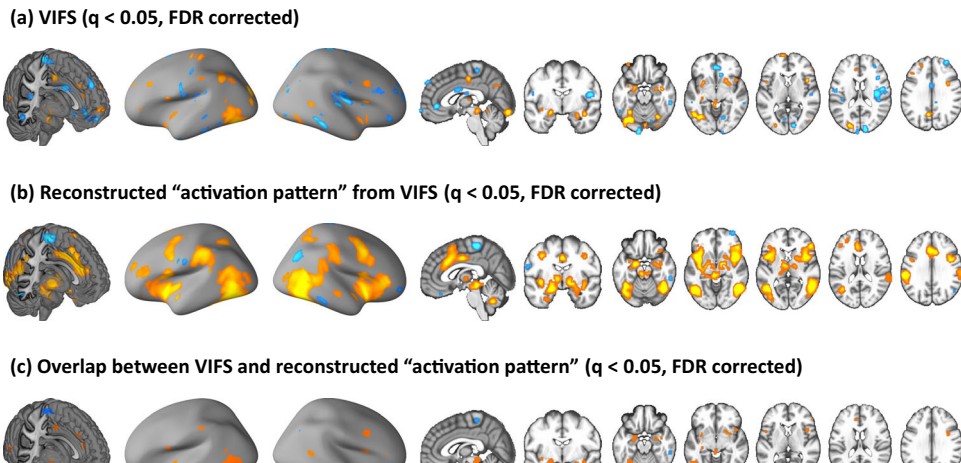

**(a) VIFS (q < 0.05, FDR corrected)**

**(b) Reconstructed "activation pattern" from VIFS (q < 0.05, FDR corrected)**

**(c) Overlap between VIFS and reconstructed "activation pattern" (q < 0.05, FDR corrected)**

**Fig. 3 Subjective experience of fear is associated with and predicted by distributed brain regions. a** shows the thresholded VIFS. **b** depicts the threshholded transformed 'activation pattern' from the VIFS. **c** shows the overlap between VIFS and transformed 'activation pattern'. All images are thresholded at $q < 0.05$, FDR corrected. Hot color indicates positive associations (**a**) or weights (**b**) whereas cold color indicates negative associations (**a**) or weights (**b**). VIFS visually induced fear signature.

**Subjective fear and conditioned defensive threat responses engage distinct neural representations in humans**. Translational fear models are strongly based on threat/fear conditioning paradigms and conditioned threat is often used synonymous with fear in the literature ('conditioned fear')[3]. However, recent fear conceptualizations emphasize potential mechanistic and neural distinctions between acquired defensive responses and the subjective experience of fear[3,6,14–16]. Against this background we examined whether the neural representation of subjective fear and conditioned threat responses were dissociable by applying the VIFS to two datasets acquired during Pavlovian threat conditioning in which an auditory cue[37] or visual cue[52] (see 'Methods' for details), respectively, was paired with a shock (CS+) while a control cue was unpaired (CS−). We specifically tested whether the VIFS generalized to discriminate CS+ versus CS−. Second, Reddan et al.[37] developed a threat-predictive signature (TPS) that accurately classified CS+ versus CS− in new individuals based on brain activity patterns. We applied the TPS to the fear paradigm data and assessed its performance in predicting subjective fear ratings by correlating the overall (between- and within-subjects) signature responses with the true ratings. We propose that if subjective fear and conditioned threat share similar neural mechanisms, the VIFS and TPS should perform well in cross-prediction: i.e., VIFS responses could predict CS+ versus CS−, and TPS responses should correlate with subjective fear ratings. Conversely, low cross-prediction indicates independence of the neural representations for 'fear' and 'conditioned threat' constructs (for similar approaches see e.g., refs. [35,36,42,53]). As shown in Fig. 5 the VIFS did not classify CS+ from CS− above chance during auditory (accuracy = 57 ± 6.0%, Cohen's $d = 0.09$, permutation test one-tailed $P = 0.234$, see also Fig. 5a) or visual threat conditioning (accuracy = 62 ± 6.4%, Cohen's $d = 0.35$, permutation test one-tailed $P = 0.265$). Given that the CS+ presentation induces higher autonomic arousal (as e.g. measured by skin conductance responses[54] in the visual threat conditioning dataset, see Supplementary Methods for details), these findings additionally suggest that the VIFS is not sensitive to general emotional arousal per se. Whereas the TPS predicted visual CS+ versus CS− cues with high accuracy (accuracy = 93 ± 3.3%, Cohen's $d = 1.30$, permutation test one-tailed $P = 0.003$) in the visual threat conditioning data, it did not predict fear ratings in our discovery, validation, or generalization cohorts (discovery: $r = 0.16$, permutation test one-tailed $P = 0.085$, Fig. 5b; validation: $r = 0.18$,

permutation test one-tailed $P = 0.111$; generalization: $r = 0.24$, permutation test one-tailed $P = 0.183$).

In support of separable brain representations underlying subjective fear experience and defensive responses towards acquired threat signals we additionally found that the VIFS and TPS pattern weights were spatially uncorrelated on the whole-brain level ($r = 0.02$, permutation test one-tailed $P = 0.125$). Moreover, we explored the joint distribution of normalized (z-scored) voxel weights of these two patterns by plotting VIFS on the $y$ axis and the TPS on the $x$ axis (for a similar approach see ref. [55]). As visualized in Fig. 5c, stronger weights across the whole-brain (sum of squared distances to the origin [SSDO]) were actually observed in the nonshared Octants (1, 3, 5, 7). Overall these results suggest distinct neural representations for subjective fear experience and conditioned threat responses. These findings provide the first evidence for separable whole-brain fMRI multivariate patterns for subjective experience of fear and conditioned threat, indicating functionally independent neural representations for subjective fear and conditioned threat.

In addition to whole-brain models, we re-trained subjective fear and conditioned threat patterns using data within integrative regions traditionally related to 'fear' but independent of sensory modality. To this end the automated meta-analysis toolbox Neurosynth[56] was used to a create a mask based on a meta-analysis of previous studies that frequently use the word 'fear'. The mask included regions (e.g., amygdala, vmPFC, aMCC, PAG and insula) showing consistent associations with 'fear' across 363 published studies (i.e., 'reverse inference'; thresholded at $q < 0.05$, FDR corrected). We found that the fear pattern trained on a priori 'fear' regions could significantly predict subjective feelings of fear (prediction−outcome $rs > 0.30$, $Ps < 0.002$ for discovery, validation, and generalization cohorts) and the threat pattern could classify unreinforced CS+ versus CS− (accuracies > 62%, $Ps < 0.008$ for auditory and visual conditioning datasets) although the performances were substantially worse as compared to whole-brain models. In support of the whole-brain findings, the two patterns were spatially not correlated ($r < 0.01$) and the conditioned threat pattern could not predict subjective fear ($rs < 0.15$, $Ps > 0.13$) and the subjective fear pattern did not distinguish unreinforced CS+ vs CS− (accuracies < 53%, $Ps > 0.69$). Together, these findings further emphasize that subjective fear- and conditioned threat-related representations within core 'fear' regions are coded by separable neural representations in humans.

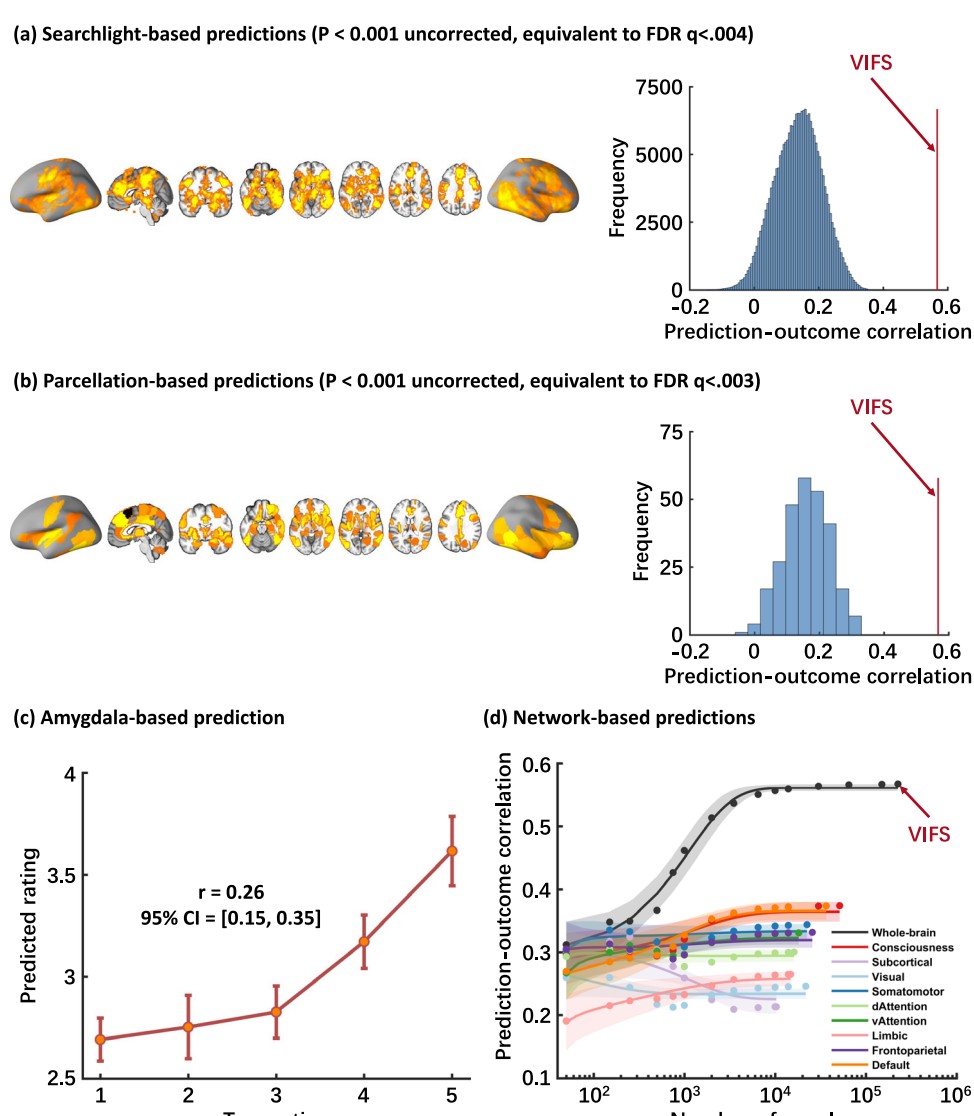

**Fig. 4 Local brain region and network predictions. a** shows brain regions that can significantly ($P < 0.001$ uncorrected, equivalent to FDR $q < 0.004$; two-sided) predict subjective fear ratings revealed by searchlight analysis. Histograms: cross-validated predictions (correlations) from local searchlight analysis. Red line indicates the prediction-outcome correlation from VIFS. **b** depicts brain regions which can significantly ($P < 0.001$ uncorrected, equivalent to FDR $q < 0.003$; two-sided) predict subjective fear revealed by parcellation-based analysis. Histograms: cross-validated predictions (correlations) from parcellations. Red line indicates the prediction-outcome correlation from VIFS. **c** shows cross-validated predictions (mean ± SE) from amygdala-based prediction analysis. Error bar indicates standard error of the mean; $r$ indicates overall (between- and within-subjects; i.e., $n = 333$ pairs) prediction-outcome Pearson correlation coefficient. **d** demonstrates that the information about subjective experience of fear is distributed across multiple systems. Model performance was evaluated as increasing numbers of voxels/features ($x$ axis) were used to predict subjective fear in different regions of interest including the entire brain (black), consciousness network (red), subcortical regions (light purple) or large-scale resting-state networks. The $y$ axis denotes the cross-validated prediction-outcome correlation. Colored dots indicate the mean correlation coefficients, solid lines indicate the mean parametric fit and shaded regions indicate standard deviation. Model performance is optimized when approximately 10,000 voxels are randomly sampled across the whole-brain. VIFS visually induced fear signature. Source data are provided as a Source Data file.

**VIFS responses mediate subjective fear induced by negative emotion.** 'Fear' is a highly aversive subjective state and represents a construct within the negative valence systems domain in the Research Domain Criteria (RDoC) matrix[57]. To separate fear from general negative affect we next investigated the spatial and functional similarities between the VIFS and PINES (picture-induced negative emotion signature) which was developed to track general negative emotion experience[32]. We found that these two signatures exhibited a weak positive spatial correlation ($r = 0.08$, permutation test one-tailed $P < 0.001$) and the VIFS was more sensitive to predict subjective fear rather than general negative emotion while the PINES more accurately predicted

general negative emotion as compared to fear (Fig. 6a; Table 2; see Supplementary results for more details).

Given that the experience of fear can be considered as a prototypical example of a negative emotion and that the PINES could, to some extent, predict subjective fear (discovery cohort: $r = 0.38$; validation cohort: $r = 0.37$) we next applied multilevel mediation analysis, which tested whether a covariance between two variables ($X$ and $Y$) can be explained by a third variable ($M$), to investigate the relation between PINES response, VIFS response and subjective fear rating. We employed two models to test (1) whether the VIFS response (mediator $M$), which measured subjective fear-specific response, could explain the

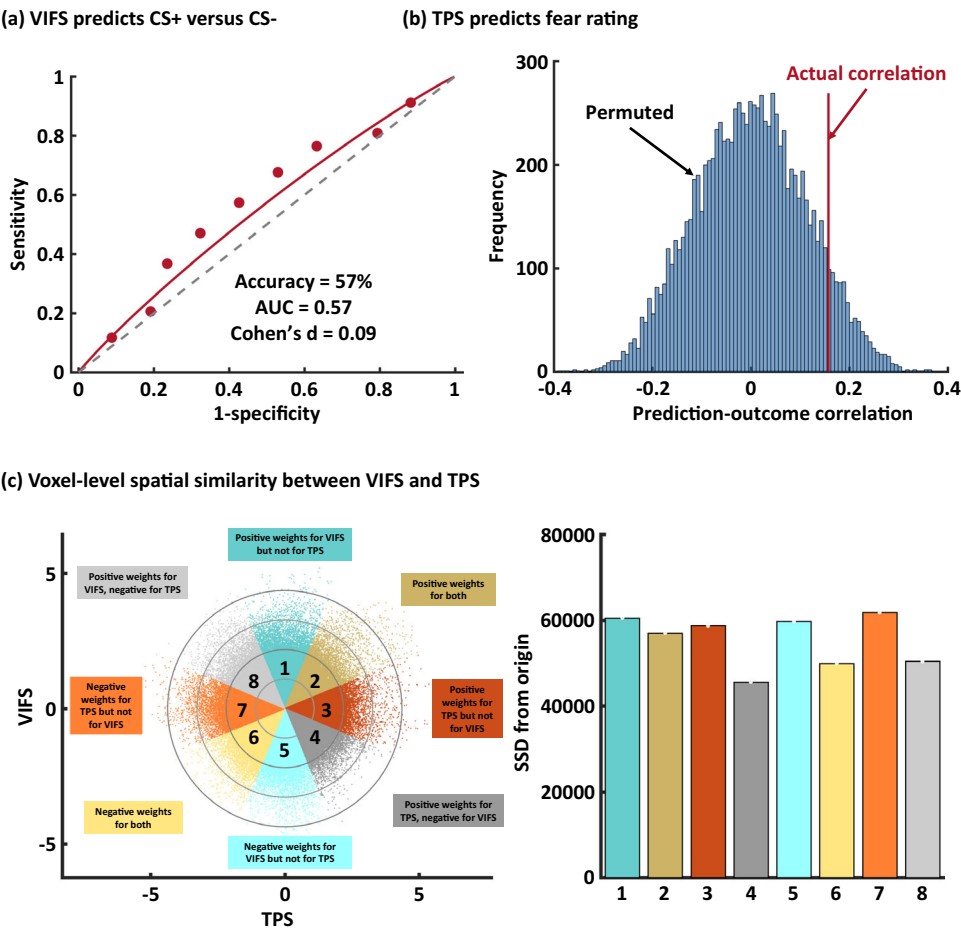

**Fig. 5 Comparing fear- and threat-predictive signatures. a** depicts that visually induced fear signature (VIFS) does not distinguish unreinforced CS+ versus CS−. **b** shows the histograms of prediction of threat-predictive signature (TPS) on fear data from nonparametric permutation test. Histograms show the distribution of null-hypothesis prediction-outcome correlations, and the red line shows the actual correlation coefficient. **c** demonstrates the scatter plot displaying normalized voxel weights for VIFS (*y* axis) and TPS (*x* axis). Bars on the right represent the sum of squared distances from the origin (0,0) for each Octant. Different colors are assigned to the eight Octants that reflect voxels of shared positive or shared negative weights (Octants 2 and 6, respectively), selectively positive weights for the VIFS (Octant 1) or for TPS (Octant 3), selectively negative weights for the VIFS (Octant 5) or TPS (Octant 7), and voxels with opposite weights for the two neural signatures (Octants 4 and 8). Source data are provided as a Source Data file.

association between nonspecific general negative emotion response (i.e., the PINES response; $X$) and fear ratings ($Y$) (our main hypothesis), and (2) as well as the alternative hypothesis of whether the general negative emotion response ($M$), which might represent the overarching emotional state of fear, mediates the association between fear signature ($X$) and fearful rating ($Y$). We found that the first model (Fig. 6b) accounted better for our data than the second one (Fig. 6c) in terms of effect size (model 1: Cohen's $d = 0.21$; model 2: Cohen's $d = 0.06$) although only a partial mediation effect was found as well as the observation that the first model worked in both discovery and validation cohorts whereas the second model only worked in discovery cohort (see Supplementary Results for more details). Our findings thus suggest that general negative emotion might not fully directly elicit subjective feeling of fear, and the response of the subjective fear neural signature could partially explain the association between negative emotion response and subjective fear rating.

**Specificity of the VIFS for the experience of fear.** Given that emotional stimuli such as the pictures we used can induce a complex array of negative emotional experiences (e.g. disgust, anger, nonspecific negative arousal), we further explored whether

the VIFS is most closely related to subjective fear. To this end we acquired ratings of associated negative emotions (disgust, anger and sadness) and emotional valence and arousal for the stimuli in an independent sample of participants ($n = 120$). The ratings were acquired online and each participant rated all stimuli with respect to one emotion ($n = 20$ subjects per emotion). Ratings were provided on a 5-point rating scale ranging from '1' (not at all) to '5' (extremely) for all dimensions except for valence which was rated from '1' (extremely positive) to '9' (extremely negative) with '5' indicating neutral.

To determine whether and to which extent the VIFS reacts to other emotional domains, we correlated the image-by-image series of normative ratings with the image-by-image variation in VIFS responses, for each emotion category assessed. Specifically, we used the single-trial beta maps for each picture and averaged the cross-validated VIFS responses for each picture. We next correlated the picture-specific group-average VIFS responses with the picture-specific group-average ratings for each emotional domain separately (for a similar approach see ref. [58]). The VIFS response was more strongly correlated with subjective fear ($r_{78} = 0.77$) than any other emotional rating (disgust: $r_{78} = 0.64$; anger: $r_{78} = 0.63$; sadness: $r_{78} = 0.60$; arousal: $r_{78} = 0.66$; valence: $r_{78} = 0.65$) suggesting that the VIFS indeed reacts most strongly

**(a) VIFS and PINES predict high versus low fear and high versus low general negative emotion separately**

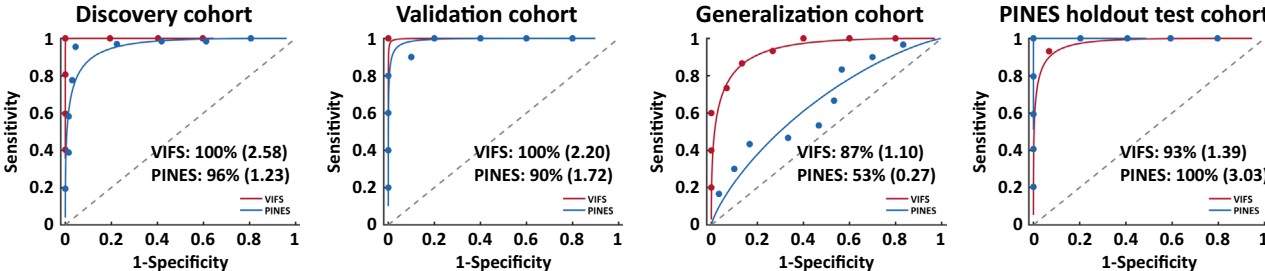

**(b) VIFS response (partially) mediates the association between PINES response and fear rating**

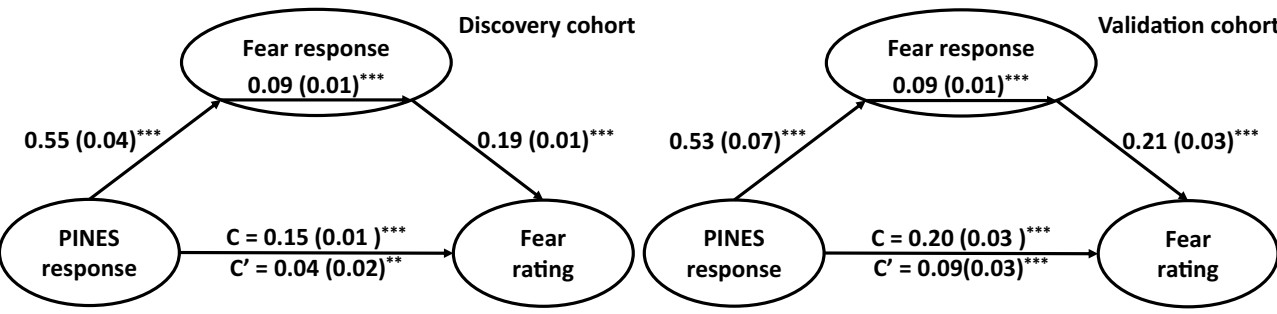

**(c) PINES response unlikely mediates PINES response – fear rating association**

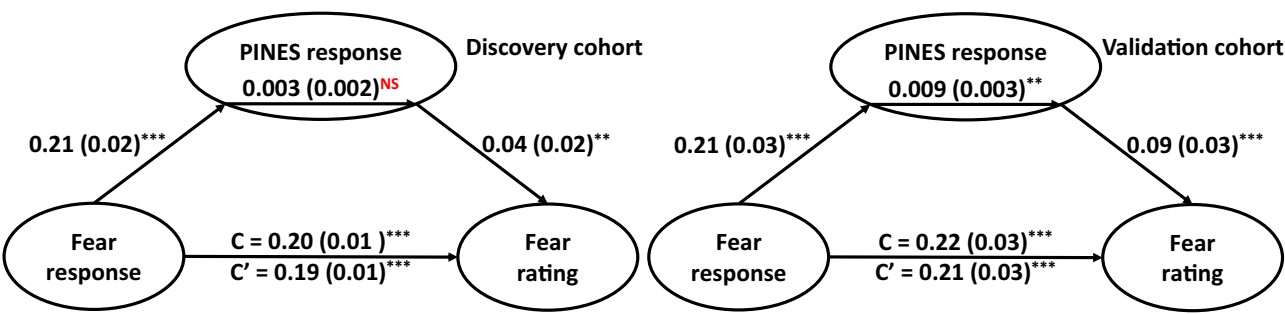

**Fig. 6 Comparing VIFS and PINES responses. a** depicts that VIFS more accurately (shown as forced-choice classification accuracy and Cohen's *d*) predicts high versus low subjective fear while PINES is more sensitive to distinguish high versus low general negative emotion. **b** shows the multilevel mediation analysis results showing that VIFS response mediates the PINES response—fear rating association in both discovery and validation cohorts. **c** shows that the PINES response does not mediate the VIFS response—fear rating association in the discovery cohort. Although the mediation effect is significant in the validation cohort, the effect size (Cohen's *d* = 0.06) is very small. ** indicates $P < 0.01$, ***$P < 0.001$, $^{NS}P > 0.5$ (bootstrap tests; two-sided; uncorrected). VIFS visually induced fear signature, PINES picture-induced negative emotion signature. Source data are provided as a Source Data file.

| Table 2 Comparing prediction (correlation) of VIFS and PINES. | | |
|---|---|---|
| **Datasets** | **VIFS** | **PINES** |
| Discovery | 0.57 [0.49, 0.63]; 0.89 ± 0.01[a] | 0.38 [0.28, 0.47]; 0.59 ± 0.04 |
| Validation | 0.59 [0.48, 0.69]; 0.87 ± 0.02 | 0.37 [0.21, 0.51]; 0.61 ± 0.07 |
| Generalization | 0.56 [0.45, 0.64]; 0.65 ± 0.06 | 0.20 [0.02, 0.36]; 0.05 ± 0.13 |
| PINES holdout | 0.29 [0.17, 0.38]; 0.63 ± 0.04 | 0.72 [0.65, 0.77]; 0.90 ± 0.01 |

We applied the VIFS and PINES to subjective fear and general negative emotion holdout datasets and calculated the overall (bootstrapped 95% CI) as well as within-subject (mean ± SE) prediction —outcome correlations between the pattern expressions and the true ratings. Source data are provided as a Source Data file.
[a]indicates cross-validated.

to subjective fear and to a lesser extent to other related negative emotions or general emotional features such as arousal. Moreover, direct comparisons of the correlations between VIFS and emotion ratings supported this conclusion and revealed significantly stronger correlations with fear than other emotions. For each subject in the discovery cohort, we correlated the cross-validated VIFS response for each picture with the picture-specific

group-average ratings for each emotional domain separately. We found that the VIFS tracked subjective fear ratings significantly better that any of the other five emotions collected in the online sample (e.g. fear versus the second best prediction, arousal; paired *t* test $t_{66} = 7.31$, $P < 0.001$, Bonforroni corrected). Together with our previous findings showing that (1) the VIFS could not distinguish CS+ (which induces higher autonomic responses as

reflected in elevated SCR responses) from CS− and (2) the prediction accuracy of VIFS on high arousing nonspecific negative emotion was substantially lower than the prediction accuracies of the subjective fear, these findings suggest that the VIFS shows reasonable specificity for subjective fear, but to some extent also captures aspects of other negative emotions or arousal which are inherently associated with fear.

To test whether the low-level visual properties of the stimuli contributed to the prediction performance, we determined several visual features of the stimuli and tested whether these can be accurately predicted by the VIFS. In detail, we measured the edge intensity (MATLAB's Canny edge detector), the saliency (http://www.saliencytoolbox.net/) as well as the visual clutters (feature congestion and subband entropy[59]) for each picture. Next, we ran similar correlational analyses as we introduced before. We found that the group-average VIFS responses were not significantly correlated with any of the visual features (most significant $r = -0.19$, $P = 0.09$). Moreover, the VIFS tracked ratings of subjective fear from the online sample significantly stronger than it tracked any of the visual features (fear versus the next closest feature, edge intensity; paired $t$ test $t_{66} = 22.15$, $P < 0.001$, Bonferroni corrected). Taken together, our findings suggest that the prediction performance was not driven by the visual properties of the stimuli.

## Discussion

In the current study we developed and validated a sensitive and generalizable brain signature for the subjective experience of fear, which predicted momentary fear on a population and individual level and thus could have potential for translational applications aiming at yielding information about individual fear experience. Furthermore, we challenge the notion that subjective fear is a product of a single brain region or network and propose that subjective fear is encoded in brain regions that span multiple neural systems. Across a series of analyses subjective fear was both associated with and predicted by distributed brain systems and fear prediction by isolated brain systems was substantial lower compared to the whole-brain approach. Driven by recent debates on whether subjective fear and the defensive response elicited by conditioned threat involve different brain circuits[3,6,14–16] we, moreover, employed a fine-grained analysis technique (MVPA) to show distinct neural representations underlying these two mental processes on the whole-brain level and in traditional 'fear' modules such as the amygdala. Finally, neural representations of subjective fear and general negative emotion exhibited shared yet separable representations, with the VIFS response mediating the association between the general negative emotion response and subjective fear. Together our findings shed light both on how subjective experience of fear is represented in the human brain and how this neural representation is separable from conditioned defensive responses and general negative emotion, respectively.

Machine-learning techniques have been increasingly used to develop integrated predictive models of activation across multiple brain systems to predict mental processes with large effect sizes (or explained variance)[29,31]. Applying support vector regression, we developed and validated a sensitive and robust whole-brain signature (VIFS) that accurately predicted the intensity of subjective fear experience across different fear induction paradigms and MRI systems. The identification of this intermediate neural signature of subjective fear is pivotal, as it may (1) provide objective neurobiological measures that can supplement self-report which can be biased by self-reflection or communicative intentions[60] and (2) promote the development and evaluation of process-specific interventions that target subjective fear experience.

Our findings have theoretical implications for ongoing debates about the neural circuits of fear, specifically the neural representation of subjective fear experience (for an overview see ref.[1]). For instance, the subcortical fear system theory suggests that feelings of fear arise from highly conserved amygdala-centered subcortical circuits[8,10,13], while high-order perspectives emphasize the contribution of the fronto-parietal 'consciousness network' to fear experience and propose that subcortical circuits are not directly responsible for fear experience[3,6,16,40]. Although limitations of the structure-centric view are widely acknowledged[44] and appraisal[43] and constructionist[2] theories have suggested that fear experience results from interactions between multiple processes and brain systems, a systematic empirical comparison of structure-centric versus distributed representation models of subjective fear was previously lacking.

The present findings challenge the structure-centric and network-centric models of subjective fear by demonstrating that subjective fear is represented in distributed brain regions, including but not limited to amygdala, prefrontal, subcortical and sensory cortices. Whereas previous predictive models focused on identifying brain regions that reliably contributed to the model for interpretation purpose[32,34] we updated and extended the characterization of the predictive model. We divided voxels into different classes based on the combination of predictive weights and reconstructed 'activation patterns'[45] and revealed that distributed brain regions, which exhibited significant predictive weights and reconstructed 'activations', contribute to both, predictions of and associations with the outcome. Second, we demonstrated that isolated regions (e.g., amygdala) and networks (e.g., 'consciousness network') predicted subjective fear to a substantially lower extent than the whole-brain signature (VIFS). Finally, around 10,000 voxels that were randomly sampled across the whole-brain could achieve high performance of predicting subjective fear, which could also be generalized to new data collected with different paradigms. Together our findings suggest that the fear circuits identified in previous studies may only represent aspects of the subjective fear experience, as reflected by comparably low effect sizes, yet that the subjective feeling of fear requires engagement of distributed brain systems. Our findings are consistent with recent MVPA studies demonstrating that affective processes including general negative emotion[32], vicarious[35] and self-experienced pain[31] are distributed across regions, and meta-analytic evidence suggesting that emotional experience is constructed in a set of highly interacting brain regions[44].

The RDoC matrix suggests several paradigms to study acute threat (or 'fear'), including fear conditioning and exposure to emotional evocative stimuli. Indeed, subjective feelings of fear and conditioned threat exhibit a pattern of similar brain activation particularly in subcortical and prefrontal cortices[24–26,37,52]. However, recent conceptualizations propose that due to the fact that the conditioned automatic defensive response represents an innate, fixed action pattern which does not necessarily require consciousness (as opposed to subjective fear which is a conscious experience), the underlying neural mechanisms might be distinct[3,6,14–16]. Traditional univariate activation analyses lack anatomical specificity and thus cannot determine whether the neural representations of overlapping activations are similar or distinct[36,61], while the MVPA approach can extract information at finer spatial scales[29,33] and permits support for or rejection of claims about neural mechanisms that are shared between mental processes[62]. Our findings indicate separable neural representations of subjective fear and the conditioned defensive response not only on the whole-brain level but also in 'core fear regions'. Our study supports and extends current conceptualizations of neurobiologically orthogonal processes and implies that

conditioned threat and subjective fear are distinct constructs within the negative valence system.

In line with the RDoC matrix suggesting that fear is a construct of the negative valence systems, the VIFS shared similar yet different characteristic functions with the PINES which tracks general negative emotional responses including sadness, anger, disgust and fear[32]. The VIFS was more sensitive to predict subjective fear as compared to other emotional domains including disgust, anger, sadness, arousal and negative valence, together with the observation that VIFS failed to predicted conditioned threat versus safety signals and VIFS responses mediated the association between PINES responses and fear ratings, suggesting that the VIFS is a more robust and specific brain marker for subjective feelings of fear.

A previous study[40], which aimed at comparing the neural representations of subjective fear and the physiological threat response, developed a decoder predictive of reported fear (i.e., AFSS) as assessed by offline ratings to different animal categories. The VIFS generalized well to the dataset used to train the AFSS, but the AFSS did not generalize to the same extend to the datasets used to train the VIFS. This might be due to the fact that the AFSS reflects more "stable" fear schemas (e.g. the general fear of spiders) while the VIFS is more accurate to capture fear across different stimulus classes and contexts. Nevertheless, both studies have consistently demonstrated that activation patterns in distributed brain systems including e.g., prefrontal regions, insula and hippocampus are predictive of subjective fear and that activation in the 'fear center' (i.e., amygdala) is not sufficient to represent subjective fear experience. These findings are consistent with a growing number of studies demonstrating that brain activation in isolated regions or specific subsystem is neither necessary nor sufficient for predicting subjective emotional experiences (see e.g. also findings from a "virtual lesion" approach in ref. [32]). Moreover, in line with our findings showing that subjective fear and conditioned threat responses engage distinct neural representations in humans Taschereau-Dumouchel et al.[40] reported distinguishable neural representations of subjective fear and its physiological correlates, together suggesting separable neural representations of subjective fear experience and hard-wired defensive responses.

The present study used IAPS-type static images as stimuli. Although ratings revealed that these images could elicit a relative robust range of subjective fear experience, the types of variations in stimuli that lead to distinct vs. similar neural encoding are still not well understood. It is for instance conceivable that video stimuli could activate the VIFS in proportion to the fear-inducing properties of the videos, or it is possible that the brain encodes dynamic stimuli differently. These possibilities could be tested in future studies. In addition, in the current study we used a SVR model to develop the VIFS and to explore the neural basis of subjective fear, however, the prediction accuracy and the contributing brain regions could be further explored by means of other candidate techniques such as SOS-LASSO which imposes a prior that the neural pattern should be sparse but also locally structured[63]. Moreover, the amygdala is often considered to be a 'fear center' or 'threat center' in animal models (for a critical discussion on the role of the amygdala in fear and threat (see also ref. [6]). Although a direct translation of threat-related neural representations in rodents to human emotional experiences is limited, a number of human lesion studies in patients with complete bilateral amygdala lesions underscores the complex role of the amygdala in fear processing in humans. In line with the 'fear center' perspective, an early human lesion study showed that a patient with focal bilateral amygdala lesions never endorsed feeling more than minimal levels of fear[20]. However, other studies in patients with bilateral focal and complete amygdala lesions

demonstrated that the amygdala was not critically required to experience panic triggered by a $CO_2$ challenge[23], subjective affective experience[64] or the modulation of the acoustic startle reflex by fear-inducing background stimuli[22], which together raise the question of whether the amygdala is causally necessary and sufficient for the experience of subjective fear in humans (for an in-depth discussion, see also ref. [6]). Whereas our findings indicate that the amygdala per se is not sufficient to represent subjective experience of fear in humans, the question of a causal role of the amygdala in subjective fear in humans cannot be ultimately addressed in the present study given the indirect nature of fMRI measurements and lack of direct experimental manipulations of the brain. In addition to a widely distributed pattern of activity, voxels in the amygdala were identified across our analytic approaches, suggesting that the amygdala may represent a part of a larger network for initiating or integrating a coordinated fear and threat response on different levels (see e.g., ref. [65]).

Pre-registration has been increasingly advocated in the field of neuroimaging prediction studies (see e.g., also recent recommendations by Poldrack et al.[66]) and might help to reduce analytic flexibility in neuroimaging analyses[67]. The analytic protocols for the present study have been established in our previous studies see e.g., refs. [32,34,68] and only the single final model was tested on the validation and generalization datasets, however, pre-registration in future studies could further facilitate analytic rigor. Moreover, in the current study we showed that subjective fear and nonspecific negative emotion shared common yet also distinct neural representations. Our findings are based on cross-prediction models and training joint-models over the emotional domains in datasets that have been acquired with matched paradigms and on an identical MRI system may help to further determine common and separable neural representations between fear experience and other emotional domains. Moreover, although we identified distinct neural representations for subjective fear and conditioned threat on the whole-brain level the corresponding decoders were developed based on studies employing different paradigms and stimuli. The independence of common neurofunctional representations of subjective fear and conditioned threat thus needs to be further evaluated. Future studies could, e.g., align the paradigms by using categorical stimuli across the paradigms (e.g., high fear vs. neutral stimuli) to further explore whether subjective fear and conditioned threat share common neural representations, particularly in local regions. However, the specificity of the shared neural basis (if one is found) to threat- and fear-related processes of interest would require further testing.

In conclusion, we identified a whole-brain neural representation for the subjective experience of fear. This visually induced fear signature was validated and generalized across participants, paradigms and fMRI scanners. Our findings demonstrate the neural basis of subjective fear is not represented by isolated brain regions or networks but instead best captured by activations in distributed regions spanning multiple brain systems. The specificity of the fear signature was further tested with conditioned defensive responses and general negative emotion experience. Our work may provide objective neurobiological measures that can supplement self-report fear and potentially be used as intermediate markers for treatment discovery that target pathological fear.

## Methods
**Participants in the discovery cohort**. Seventy healthy, right-handed participants were recruited from the University of Electronic Science and Technology of China in this study. Exclusion criteria included color blindness; current or regular substance or medication use; current or history of medical or psychiatric disorders; any

contraindications for MRI. Due to the excessive head motion (>1 voxel) during fMRI scanning data from three participants were excluded, leading to a final sample of $n = 67$ participants (34 females; mean ± SD age = 21.5 ± 2.1 years). All participants provided written informed consent, and the study was approved by the local ethics committee at the University of Electronic Science and Technology of China and was in accordance with the most recent revision of the Declaration of Helsinki. After the experiment, participants were compensated 80 RMB for participation.

**Stimuli and paradigm used in the discovery cohort.** The fear rating task consisted of 4 runs with each run encompassing 20 photographs (including humans, animals, and scenes) from the IAPS (International Affective Picture System), NAPS (Nencki Affective Picture System), and internet. A total of 80 stimuli was employed, with each presented once. Stimuli were presented using the E-Prime software (Version 2.0; Psychology Software Tools, Sharpsburg, PA). Participants were instructed to pay attention to the pictures when they came on the screen. Each trial consisted of a 6 s presentation of the picture followed by a 2 s fixation-cross separating the stimuli from the rating period. Participants then had 4 s to report the fearful state they experienced for the stimuli using a 5-point Likert scale where 1 indicated neutral/slightest fear and 5 indicated most strongly fear. Finally, there was a 6-s rest period (fixation-cross) before the presentation of the next picture (Fig. 1a). All of the subjects reported rating '1−4' in their responses while 2 out of 67 subjects did not use rating '5'.

**Discovery cohort MRI data acquisition and preprocessing.** MRI data were collected on a 3.0-T GE Discovery MR750 system (General Electric Medical System, Milwaukee, WI, USA) (see Supplementary Methods for details). Functional MRI data were preprocessed using Statistical Parametric Mapping (SPM12 v7487, https://www.fil.ion.ucl.ac.uk/spm/software/spm12/). The first five volumes of each run were discarded to allow MRI T1 equilibration. Prior to preprocessing of functional data, image intensity outliers resulting from gradient and motion-related artefacts were identified using CanlabCore tools (https://github.com/canlab/CanlabCore) based on meeting any of the following criteria: (a) signal intensity >3 standard deviations from the global mean or (b) signal intensity and Mahalanobis distances >10 mean absolute deviations based on moving averages with a full-width at half maximum (FWHM) of 20 image kernels. Each time-point identified as outliers was included as a separate nuisance covariate in the first-level model. Then, functional images were corrected for differences in the acquisition timing of each slice and spatially realigned to the first volume and unwarped to correct for nonlinear distortions related to head motion or magnetic field inhomogeneity. The anatomical image was segmented into gray matter, white matter, cerebrospinal fluid, bone, fat and air by registering tissue types to tissue probability maps. Next, the skull-stripped and bias-corrected structural image was generated and the functional images were co-registered to this image. The functional images were subsequently normalized the Montreal Neurological Institute (MNI) space (interpolated to $2 \times 2 \times 2$ mm voxel size) by applying the forward deformation parameters that were obtained from the segmentation procedure, and spatially smoothed using an 8-mm full-width at half maximum (FWHM) Gaussian kernel.

**First-level fMRI analysis used in the discovery cohort.** We conducted two separate subject-level GLM (general linear model) analyses. The first GLM model was used to obtain beta images for the prediction analysis. In this model we included five separate boxcar regressors time-logged to the presentations of pictures in each rating (i.e., 1−5), which allowed us to model brain activity in response to each fear level separately. To model any effects related to motor activity the model also included one boxcar regressor indicating the rating period. The fixation-cross epoch served as implicit baseline. The second GLM model included two regressors of interest, with one modeling the picture viewing period and the other modeling the fear rating period. Additionally, the design matrix included fear ratings (1−5) reported for each picture as a parametric modulator for the picture viewing period.

All task regressors were convolved with the canonical HRF function and a high-pass filter of 128 s was applied to remove low frequency drifts. Time series from multiple runs were concatenated using SPM's spm_fmri_concatenate.m function, which included an intercept for each run and corrected the high-pass filter and temporal non-sphericity calculations. Regressors of non-interest (nuisance variables) included (1) six head movement parameters and their squares, their derivatives and squared derivatives (leading to 24 motion-related nuisance regressors in total); and (2) indicator vectors for outlier time points (see above for details).

**Participants in the validation cohort.** Twenty-two healthy, right-handed participants were recruited from the University of Electronic Science and Technology of China in this study. Due to excessive head motion (>1 voxel) during fMRI scanning data from two participants were excluded leading to a final sample of $n = 20$ participants (6 females; mean ± SD age = 21.75 ± 2.61 years). All participants provided written informed consent, and the study was approved by the local ethics committee at the University of Electronic Science and Technology of China and

was in accordance with the most recent revision of the Declaration of Helsinki. After the experiment, participants were compensated 90 RMB for participation.

**Stimuli and paradigm used in the validation cohort.** The fear rating task consisted of 2 runs with each run encompassing 30 photographs (60 in total) from the IAPS, NAPS, and internet. Fifty-eight out of 60 stimuli were overlapped with the stimuli used in the discovery cohort. Stimuli were presented using the E-Prime software. Participants were instructed to pay attention to the pictures when they came on the screen. Each trial consisted of a 6 s presentation of the picture followed by a jittered fixation-cross (1, 2 or 3 s). Participants then had 4 s to report the emotional state they experienced for the stimuli using a 5-point Likert scale where 1 indicated minimal fear/neutral and 5 indicated very strong fear. Finally, there was a jittered fixation-cross epoch (4, 5, or 6 s) before the presentation of the next picture (Fig. 1b). All of the subjects reported rating '1−5' in their responses.

**Validation cohort MRI data acquisition, preprocessing, and first-level fMRI analysis.** Imaging data acquisition, preprocessing, and subject-level GLM analysis were identical to the discovery cohort.

**Generalization cohort.** The details of the generalization cohort were reported in previous studies[40]. Briefly, 31 participants (15 females; mean ± SD age = 23.29 ± 4.21 years) underwent a 1 h fMRI session (see Supplementary Information for the MRI acquisition details) where they were presented with 3600 images consisting of 30 animal categories and 10 object categories (90 different images per category). The stimuli were grouped in blocks of 2, 3, 4 or 6 images of the same category with each stimulus presented for 1 s (no interblock or interstimulus interval). Subjective fear ratings (0 = 'no fear' to 5 = 'very high fear') for each category were established before the fMRI procedure without presenting any fearful stimuli. We used labels 1–6 instead of 0–5 in Fig. 2f, g for display purposes (of note, this procedure changes only the intercept/bias but not the pattern weights of the predictive model and has no effects on the prediction−outcome correlation or the forced-choice classification). The least-square separate single-trial analysis approach was employed to iteratively fit a GLM to estimate the brain response to the first image of each block and then the within-subject beta images with the same fear ratings were averaged, which resulted in one beta map per rating for each subject (for paradigm, MRI acquisition and analysis details see ref. [40]).

**Multivariate pattern analysis.** We applied whole-brain (restricted to a gray matter mask[36,69]) multivariate machine-learning pattern analysis to obtain a pattern of brain activity that best predicted participants' self-reported fear ratings. Of note, the findings were comparable with a whole-brain mask with white matter and cerebrospinal fluid included. We employed the support vector regression (SVR) algorithm using a linear kernel ($C = 1$) implemented in the Spider toolbox (http://people.kyb.tuebingen.mpg.de/spider) with individual beta maps (one per rating for each subject) as features to predict participants' fear ratings of the grouped pictures they viewed while undergoing fMRI. Of note, we only used data from the discovery cohort to develop the VIFS. To evaluate the performance of our algorithm, we used a $10 \times 10$-fold cross-validation procedure on the discovery cohort during which all participants were randomly assigned to 10 subsamples of 6 or 7 participants using MATLAB's cvpartition function. The optimal hyperplane was computed based on the multivariate pattern of 60 or 61 participants (training set) and evaluated by the excluded 7 or 6 participants (test set). This procedure was repeated ten times with each subsample being the testing set once. To avoid a potential bias of training-test splits, the cross-validation procedures throughout the study were repeated ten times by producing different splits in each repetition and the resultant prediction performance were averaged to produce a convergent estimation[36,70]. Several metrics have been proposed to evaluate the predictive power of multivariate predictive signatures (see e.g., ref. [66]); however, the advantages and disadvantages of each metric are still a matter of debate, and metrics vary subtly in their properties. To facilitate a robust determination of the predictive accuracy of the neurofunctional signature we therefore employed various metrics including correlation, RMSE, EVS and forced-choice classification accuracy. Specifically, we used overall (between- and within-subjects; 333 pairs in total) and within-subject (5 or 4 pairs per subject) Pearson correlations between the cross-validated predictions and the actual ratings to indicate the effect sizes and the RMSE and explained variance score to illustrate overall prediction error. The explained variance score was assessed using the following formula: explained variance score $= 1 - (\text{var}[y - \hat{y}]/\text{var}[y])$, where $y$ is the true rating, $\hat{y}$ is the VIFS response (plus intercept) and var indicates the variance (as implemented in software packages such as scikit-learn). In addition, we assessed classification accuracy of the VIFS using a forced-choice test, where signature responses were compared for two conditions tested within the same individual, and the higher was chosen as more fearful. We also applied the fear-predictive pattern (trained on the whole discovery cohort) to the validation and generalization cohorts to obtain a signature response for each map (that is, the dot-product of the VIFS weight map and the test image plus the intercept) to assess the prediction performance of the VIFS using a permutation test with 10,000 random shuffles. Given that the cross-validated permutation test is very time consuming the inferences on model performance were only performed using permutation testing on the validation and generalization cohorts.

**Comparing the performance of VIFS with the AFSS**. A previous study has developed a whole-brain fear decoder[40]. To compare the performance of VIFS with the AFSS we applied both patterns to the discovery, validation and generalization cohorts and assessed the overall prediction−outcome correlation as well as two-alternative forced-choice classification accuracies between low, moderate and high fear based on the pattern expressions.

**Within-subject trial-wise prediction**. Here we tested whether the VIFS could predict individual trial-by-trial subjective fear. To this end we performed a single-trial analysis, which was achieved by specifying a GLM design matrix with separate regressors for each stimulus. Each task regressor was convolved with the canonical hemodynamic response function. Nuisance regressors and high-pass filter were identical to the above GLM analyses. One important consideration for single-trial analysis is that the beta estimates for a given trial could be strongly affected by acquisition artifacts (e.g., sudden motion) that co-occur during a trial. For each subject we therefore excluded trials with variance inflation factors (a measure of design-induced uncertainty due to multicollinearity with nuisance regressors) >3 from subsequent analyses (overall ~6% trials were excluded). Next, we calculated the VIFS pattern expressions of these single-trial beta maps (i.e., the dot-product of vectorized activation images with the VIFS weights), which were finally correlated with the true ratings for each participant separately. For subjects in the discovery cohort we used the $10 \times 10$-fold cross-validation procedure to obtain the VIFS response of each single-trial beta map for each subject.

**Determining brain regions associated with and predictive of subjective fear**. To identify neural circuits underlying subjective experience of fear, we employed a series of analyses. Firstly, we performed one-sample $t$ test on the first-level univariate parametric modulation beta maps to see which brain regions' activation was associated with fear ratings. Next, we used multivariate analyses to locate brain regions that predictive of and associated with fear ratings separately as well as brain regions showing an overlapping effect. Specifically, we evaluated the consistency of each weight for every voxel in the brain across within-subject multivariate classifiers (developed with single-trial data) using a one-sample $t$ test. The thresholded map ($q < 0.05$, FDR corrected) showed the consistent fear-predictive brain regions across subjects. To this end we performed a prediction analysis (linear SVR with $C = 1$) for each subject in the discovery cohort separately using their single-trial data ($10 \times 10$-fold cross-validated) and only included participants whose fear ratings could be significantly predicted by their brain data (evaluated by prediction −outcome Pearson correlation; $n = 60$). Of note, similar results were found when including the entire sample.

Given that the predictive brain regions could be related to (in this case) fear processing as well as suppressing the noise in the data[45], we transformed the within-subject patterns to 'activation patterns' using the following formula: $A = \text{cov}(X) \times W \times \text{cov}(S)^{-1}$, where $A$ is the reconstructed activation pattern, $\text{cov}(X)$ is the covariance matrix of training data, $W$ is the pattern weight vector, and $\text{cov}(S)$ is the covariance matrix of the latent factors, which is defined as $W^T \times X$. This reconstructed activation is also similar to the 'structure coefficients' in the statistical literature. Previous studies have argued that both betas and structure coefficients are necessary to interpret the model[47]. Essentially, the beta indicates the predictive slope and direction of effect controlling for other variables in the model whereas the structure coefficients indicates the direction of the relationship between the variable and the model without controlling for other variables—i.e., in the current study, which voxels are positively and which are negatively related to the predicted subjective fear. The significant brain regions (one-sample $t$ test thresholded at $q < 0.05$, FDR corrected) exhibited the consistent fear-associative effect. In parallel with with-subject models we conducted bootstrap tests, where we took 10,000 samples with replacement from the discovery cohort, repeated the prediction process with each bootstrap sample, and calculated $Z$ scores and two-tailed uncorrected $P$ values with the mean and standard deviation of the sampling distribution, on the population-level fear-predictive pattern (i.e., the VIFS) as well as the transformed 'activation pattern' from the VIFS to identify the reliable fear predictive and associative brain regions of the VIFS (thresholded at $q < 0.05$, FDR corrected). To facilitate the determination and interpretation of a subjective fear signature convergent univariate and multivariate approaches were implemented. Spatial patterns (or regions) that were consistently observed across backward and forward models were considered as reliably and consistently associated and predictive of subjective fear.

Furthermore, we asked whether fear processing could be reducible to activations in a single brain region (e.g., amygdala) or network (e.g., subcortical regions). To examine this hypothesis, we employed whole-brain searchlight (three-voxel radius spheres)[71]—and parcellation (274 cortical and subcortical regions)[72]—based analyses to identify local regions predictive of fear and compared model performances of local regions with the whole-brain model (i.e., the VIFS). In addition, we compared prediction performances of amygdala (based on Anatomy Toolbox version 2.2c; available online in the Cognitive and Affective Neuroscience Laboratory Github repository at https://github.com/canlab/Neuroimaging_Pattern_Masks) and large-scale networks to the whole-brain approach. The networks of interest included seven resting-state functional networks[50,51], a subcortical network (including the striatum, thalamus, hippocampus and amygdala) and a 'consciousness network' proposed by LeDoux

and Pine[6], which composed of anterior cingulate cortex, inferior frontal gyrus, middle frontal gyrus, superior frontal gyrus, orbitofrontal gyrus, rectus, olfactory and insula from the AAL atlas and the posterior parietal cortex from Shirer et al.[73]. For these analyses we trained and tested a model for each searchlight sphere, parcellation, brain region or network separately using the discovery data ($10 \times 10$-fold cross-validated).

**Comparing the similarities of the VIFS and a threat-predictive signature**. To examine the functional and spatial similarities between the VIFS and the TPS threat-predictive signature[37]; which predicts the defense responses elicited by threat conditioning, we (1) applied VIFS to distinguish unreinforced CS+ versus CS− and predicted subjective fear ratings using the TPS, and (2) examined the voxel-level spatial similarity between these two signatures. Inference on model performance was performed using permutation testing with 10,000 random shuffles. Given that the fear and threat conditioning studies employed visual and auditory cues, respectively, we next tested whether the dissociable effects based simply on differences in sensory processing by applying both signatures to an independent visual threat conditioning dataset[52]. If the predictions were sensory-dependent the TPS would not distinguish visual (unreinforced) CS+ versus CS− whereas the VIFS might predict visual CS+ from CS−. To this end we included two datasets that employed an auditory and visual threat conditioning paradigm during which a previously neutral stimulus was paired with an unpleasant shock (CS+) while another matched stimulus was not paired (see Supplementary Methods for details).

**Multilevel two-path mediation analysis**. In order to test the relationship between VIFS response, fear rating and the PINES response, we conducted two multilevel mediation analyses using the Mediation Toolbox (https://github.com/canlab/MediationToolbox). A mediation analysis tests whether the observed covariance between an independent variable ($X$) and a dependent variable ($Y$) could be explained by a third variable ($M$). Significant mediation effect is obtained when inclusion of $M$ in a path model of the effect of $X$ on $Y$ significantly alters the slope of the $X$–$Y$ relationship. That is, the difference between total (path $c$) and direct (non-mediated, path $c'$) effects of $X$ on $Y$ (i.e., $c − c'$), which could be performed by testing the significance of the product of the path coefficients of path $a \times b$, is statistically significant. The multilevel mediation analysis is designed to explain both within- and between-subject variations in the same model by treating the subject as a random effect[61]. The first-level accounts for the mediation effects within each individual participant and the second-level tests for consistency across participants and allows for population inference. In the current study we tested whether (1) VIFS response mediated the association between PINES response and fear rating and (2) PINES response mediated the relationship between VIFS response and fear rating. To this end the VIFS and PINES responses were calculated by dot-product of the single-trial beta maps with the VIFS (through cross-validation procedure for the discovery cohort) and PINES patterns respectively for each subject. We used bias-corrected accelerated bootstrapping (10,000 replacements) for significance testing.

**Reporting summary**. Further information on research design is available in the Nature Research Reporting Summary linked to this article.

## Data availability
The visually induced fear signature (VIFS), the discovery cohort that has been used to develop and test the VIFS and the thresholded statistical maps are available at https://figshare.com/articles/dataset/Subjective_fear_dataset/13271102. The data from the validation cohort are from an ongoing project and are available from the corresponding authors upon request. The data from the generalization cohort are from a previous study[40] and are available from the ATR repository (https://bicr.atr.jp/decnefpro/) and from the corresponding authors upon reasonable request. Source data are provided with this paper.

## Code availability
Data were analyzed using CANlab neuroimaging analysis tools available at https://github.com/canlab/ and from https://github.com/zhou-feng/fMRI-studies/tree/main/Fear_experience_signature.

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

## Acknowledgements

We thank Vincent Taschereau-Dumouchel and colleagues for sharing their data. This work was supported by the National Key Research and Development Program of China (Grant No. 2018YFA0701400) to B.B.; National Institute of Mental Health (R01MH076136; R01MH116026) to T.D.W.; National Natural Science Foundation of China (Grant No. 31530032) to K.M.K.; China Postdoctoral Science Foundation (Grant No. 2018M643432) to W.Z.

## Author contributions

F.Z., T.D.W. and B.B. designed the experiment, analyzed the data and drafted the manuscript. F.Z., W.Z., Z.Q., and Y.G. conducted the experiment. W.Z., S.Y. and K.M.K. provided feedback and revised the manuscript.

## Competing interests

The authors declare no competing interests.
