## [Peer Review File · Nature Communications]

A distributed fMRI-based signature for the subjective experience of fearREVIEWER COMMENTS

Reviewer #1 (Remarks to the Author):

General points:

This is an important study that adds to the growing acknowledgment that fear is not simply a state programmed in amygdala activity. However, I have several comments.

The main claim of the study is that the neural code of fear is widely distributed in the brain. Although this is a correlational finding, the authors should be more explicit in noting that although voxels associated with the prediction of fear are present in many brain areas, doesn't necessarily mean that the necessary and sufficient voxels are also distributed all over the brain. If one were to extend their approach and include facial expressions for instance, it would probably be possible to train an even better decoder. But that wouldn't mean that the facial expression is also causally related to the subjective experience.

The study is very similar one by Tascereau-Dumouchel in Mol Psych paper in 2019 (same algorithm--Support Vector Regression). One important difference is that the present study predicts online fear ratings (i.e. following images from the IAPS) while the previous predicted offline categorical ratings. Also, the sample size is twice as large in the new study, but the conclusions were similar.

One interesting finding is that the present decoder generalizes well to the one of Tascereau-Dumouchel, but the dose not generalize well to the present one. One possibility is that the other decoder might be relying on features of more "stable" fear schemas (e.g., fear of spider) while the present decoder might be more tailored for the fear of diverse situations (e.g., someone pointing at you with a gun).

Methodological points:

This is a relatively big study for a task-based neuroimaging study. It has 2 external validation samples which is in line with current machine learning guidelines. This is actually quite important as cross-validation procedures (conducted within a single dataset) have been shown to overfit the data and to poorly estimate the true predictive power of the signature (see Varoquaux, 2018, Neuroimage). However, it seems that the authors oscillate between testing their signature on external validation samples and running cross-validated analyses. Essentially, some of the results supporting their main point (and presented in the main figures) are still conducted using cross-validation procedures (Figure 4). This could be fixed by moving some data out of the supplementary material and by running new analyses on the existing data.

Another point is that the study was not pre-registered. When I read the method section, it shows. The analytical decisions are not so obvious and the reader only has access to what came up as significant. Although this is a fairly common practice, it does not reflect the best practices to be expected from a Nature paper.

The interpretation of weights of such decoding models is not so straightforward even using their Bootstrap procedure (see Kriegeskorte and Douglas, 2019, <https://www.sciencedirect.com/science/article/abs/pii/S0959438818301004>). Essentially, high weights can be due to complex interactions with other features. As a result, most papers do not report or interpret weights.

The correlation between predicted and actual values is not the best metric to assess the predictive power of the signature. They should consider using the coefficient of determination (R^2) (see Poldrack, Huckins, & Varoquaux, 2019, JAMA <https://jamanetwork.com/journals/jamapsychiatry/article-abstract/2756204>).

The analyses presented in Figure 3 are not easy to interpret. They should present their combined map clearly and move the remaining maps (that were used to generate the combined one) to the supplementary material.

Reviewer #2 (Remarks to the Author):

Review of Zhou et al., "A distributed fMRI-based neuromarker for the subjective experience of fear"

In this study, the authors develop and validate a model that can decode (across participants) the subjective experience of fear from fMRI data. The authors also distinguish the distributed neural pattern that corresponds to subjective fear from the neural representations of conditioned threat and negative affect, and they show that the output of their neural decoder mediates the association between the general negative affect response and subjective fear. Overall, I thought this was a very nice paper; it is closely in the spirit of Wager's other work (e.g., identifying distributed neural correlates of pain ratings) and -- like that other work -- should have strong clinical utility. I don't have any major concerns, but I have listed some miscellaneous points below (none of the suggested analyses are mandatory)

line 92: "V. T. decoder" -- sorry that I missed this, but what does V. T. stand for?

line 98: "been challenged BY"

did the authors consider doing hyperparameter optimization (e.g., of the "C" parameter) through nested cross-validation?

did the authors consider using recursive feature elimination to find the most compact set of informative voxels?

another technique that might be useful for refining the pattern used by the classifier is SOS-Lasso, which imposes a prior that the neural pattern should be sparse but also locally structured; see <https://www.jneurosci.org/content/41/5/1019.abstract>

lines 171-172: clarify that the predicted ratings were averaged as a function of reported fear rating within subjects before the within-subject correlations were computed (or, if I am misunderstanding, please clarify)

line 277: I would like to see a bit more info about the forward modeling technique in the main paper, as I was unfamiliar with the particular method used here

line 774: "in parallel with withIN-subject models"

line 1062: "Panel B SUMMARIZES"

Sincerely,

Ken Norman (I sign all of my reviews)

Reviewer #3 (Remarks to the Author):

In this study, BOLD-fMRI data from multiple cohorts of subjects and protocols (and indeed scanners) were leveraged to argue for a generalizable multivoxel signature of visually induced fear. The first study used a validated set of visual stimuli that evoke parametric ratings of subjective fear experience. In 67 subjects this was used to discover the multivoxel features that could predict fear, using a linear predictive model; in addition to the cross-validated initial findings, an independent confirmation cohort (N=20) was then also used (decoder fully trained on discovery sample, tested on this independently held out sample of 20). The predictive accuracy was very high (e.g., $r > 0.8$) and dichotomizing the fear ratings to high vs. low yielded classification performance of 100%. The VIFS did not depend on occipital cortex and emerged about 4sec after image onset; it generalized to predicting fear in other datasets, even acquired on other scanners. Finally, this population-trained VIFS was applied to the trial wise beta maps in each individual

subjects, and was able to predict, on each trial, the intensity of fear experience for most subjects (r s around 0.4). Overall, this suggested a sensitive and generalizable set of neural features that a linear model could use to predict subjective fear in general and individually.

The authors next investigated the neuroanatomical substrates of their findings. Standard univariate analyses showed activations and deactivations to fear distributed over many structures, as expected. These mostly overlapped with the maps obtained from the weights of the MVPA features obtained from the above predictive models. To control for the possibility that these feature weights may not selectively reflect fear intensity (but may be predictive because they cancel out noise from other voxels, a well-known issue in interpreting the weights of regularized regressions), they also computed the structure coefficients ("forward model"). Bottom line: these all looked very similar.

Another set of approaches used searchlights or ROIs. Even when carefully comparing for the number of features, the VIFS always performed a lot better than any of these more anatomically circumscribed sets of features (although they could predict to some extent). It was found that about 10,000 voxels reached optimal predictive accuracy, provided they were sampled all over the brain.

Overall, I find this a technically impressive study. However, I have questions about the interpretation, as well as some others:

1. I wonder about the validity and specificity of the stimuli and ratings. IAPS-type images were used, which is not ideal (videos are much better). As far as I can tell, subjects were not given a choice or freely asked how they felt when they looked at these, but forced only to give a rating on fear. So how do we know how specific this is to fear? It would seem that the findings could also be just explained by arousal — could the authors please address this question. The construct of fear is not well assessed with these stimuli or questions, since no other emotion dimensions seem to have been assessed. This stands in contrast to the detailed neuroanatomical investigation to show how specific (or distributed) anatomical features are— but no equivalent dissection is undertaken for "fear" leaving the validity and specificity to fear rather understudied here. The study shows that the VIFS is predicting something, but it is far from clear that what it is predicting is specific to fear, as far as I can tell.
2. The comparison to the conditioning study is a little hard to interpret, since different objectives are used. For the present fear study, it is the parametric rating of experienced fear, irrespective of the stimuli. For the conditioning study, it was the CS+ or CS- without any fear ratings. This strikes me like comparing apples and oranges. What would be a better comparison would be to compare to actual ratings of subjective threat experience in the conditioning study.
3. Both for the comparison to the conditioning study, and for the comparison to the negative affect study, these comparisons are not ideal. In each case, a predictive model was optimized to each dataset alone, and then the features or cross-decoding were compared. But to answer the question of whether or not there are shared representations, a joint model should be trained — that is, we should ask, if we want to predict BOTH fear and negative affect, can we train such a model and how well does it do. Such a jointly trained model could well draw on features that are quite different from those in the models based on any one dataset in isolation.
4. The analyses are sophisticated and many checks have been put in place to ensure reliability. But the question of validity remains: what exactly does the VIFS predict? The authors note some of the debates in the field at the beginning of their paper. The fact that their results generalize, but are distinct from conditioned threat or negative affect, all argue that whatever is being predicted, it is something that accompanies fear. But what exactly is that? As is common, the authors refer to the VIFS as a "neural representation" of fear, but this seems far from clear, especially given the debates. I am also somewhat puzzled by the rather long latency to find the VIFS (ca. 4 seconds after stimulus onset), which would be more consistent with autonomic or motoric consequences of fear rather than fear itself. Finally, I find the extremely high predictive accuracy of the VIFS puzzling. Surely not everybody is experiencing fear the same way from these stimuli?

5. The discussion and conclusions of the paper should include a more nuanced discussion of how these findings relate to debates in the literature, and of the limitation of the present findings. A lot of the literature on fear is based on data from rodents, and it is certainly possible that the rodent brain “represents” fear differently than does the human brain. The present findings are all based on fMRI, which is indirect, has poor spatiotemporal resolution, and, most important, shows only correlations. So it could be quite possible that the amygdala is causally necessary for fear, even subjective fear in humans, even though many other brain regions carry information about subjective fear once it has been caused.

6. The actual protocol seems somewhat strange. In the discovery study, only 20 stimuli are used (in 4 runs), shown each for 6 seconds with a rating subsequently. That’s a total of about 16 minutes of scanning. I would have expected considerably more data per subject, and a larger variety of stimuli. Also, I could not find any description of the low-level visual properties of the stimuli, which needs to be controlled for. Finally, asking subjects to give explicit ratings partly confounds conceptualization required in the task with the subjective experience— it would be important to compare these data to ones where no ratings are required during the scan (but subjects rate the stimuli again outside the scanner subsequently).

Reviewer #1 (Remarks to the Author):

General points:

This is an important study that adds to the growing acknowledgment that fear is not simply a state programmed in amygdala activity. However, I have several comments. The main claim of the study is that the neural code of fear is widely distributed in the brain. Although this is a correlational finding, the authors should be more explicit in noting that although voxels associated with the prediction of fear are present in many brain areas, doesn't necessarily mean that the necessary and sufficient voxels are also distributed all over the brain. If one were to extend their approach and include facial expressions for instance, it would probably be possible to train an even better decoder. But that wouldn't mean that the facial expression is also causally related to the subjective experience. The study is very similar one by Tascereau-Dumouchel in Mol Psych paper in 2019 (same algorithm--Support Vector Regression). One important difference is that the present study predicts online fear ratings (i.e. following images from the IAPS) while the previous predicted offline categorical ratings. Also, the sample size is twice as large in the new study, but the conclusions were similar.

One interesting finding is that the present decoder generalizes well to the one of Tascereau-Dumouchel, but the dose not generalize well to the present one. One possibility is that the other decoder might be relying on features of more "stable" fear schemas (e.g., fear of spider) while the present decoder might be more tailored for the fear of diverse situations (e.g., someone pointing at you with a gun).

Methodological points:

This is a relatively big study for a task-based neuroimaging study. It has 2 external validation samples which is in line with current machine learning guidelines. This is actually quite important as cross-validation procedures (conducted within a single dataset) have been shown to overfit the data and to poorly estimate the true predictive power of the signature (see Varoquaux, 2018, Neuroimage). However, it seems that the authors oscillate between testing their signature on external validation samples and running cross-validated analyses. Essentially, some of the results supporting their main point (and presented in the main figures) are still conducted using cross-validation procedures (Figure 4). This could be fixed by moving some data out of the supplementary material and by running new analyses on the existing data.

Response: We agree with the reviewer that cross-validation might in principle over-fit the data, and in the original version we therefore tested the prediction accuracy using external independent validation/generalization datasets for all whole-brain analyses including the single trial and multi-level mediation analyses. However, as mentioned by the reviewer in some instances (searchlight- and parcellation-based analyses) only cross-validation approaches were employed. Based on the comment from the reviewer we now included predictions across datasets consistently for all analyses, i.e., we predict validation and generalization cohorts with models trained on the discovery cohort. The results and conclusions remained stable across the additional analyses. Of note, no single-trial beta maps were available for the generalization cohort, and thus for the within-subject single-

trial predictions and the multilevel mediation analysis we only included results from the discovery cohort (via cross-validation) as well as cross-dataset validation using data from the validation cohort. The new results have been included in Supplementary Fig. 4 in the revised version of the manuscript.

Another point is that the study was not pre-registered. When I read the method section, it shows. The analytical decisions are not so obvious and the reader only has access to what came up as significant. Although this is a fairly common practice, it does not reflect the best practices to be expected from a Nature paper.

Response: We agree with the reviewer on the importance of pre-registration of neuroimaging studies in general, given that analytical flexibility can have a substantial effect on the results and in turn the corresponding conclusions. In contrast to our pharmacological fMRI studies (which we have pre-registered since 2013; see e.g. Becker et al., 2013) we did not pre-register the present or our previous MVPA prediction studies, chiefly because we included holdout test sets tested only once, which are used to prevent bias and accomplish the goal of reducing analytic flexibility that pre-registration is intended to accomplish (see e.g., Chang, Gianaros, Manuck, Krishnan, & Wager, 2015; Wager et al., 2013; Zhou et al., 2020). In contrast to other research fields that advocate pre-registration, proposals to pre-register MVPA-based prediction neuroimaging studies have been put forward rather recently (e.g. Poldrack et al., 2019). In addition, flexibility in the present study is limited by the facts that (1) the general procedures, analytic protocols and workflow - i.e. determining voxel-level predictions to generate neurofunctional signatures for specific mental processes and subsequently comparing spatial and functional similarities between signatures to establish the specificity of the developed signatures - have been relatively well established in our previous studies (see e.g., Chang et al., 2015; Wager et al., 2013; Zhou et al., 2020), and (2) each analytic step aligns with testing a specific hypothesis which is rooted in recent overarching frameworks on fear and threat processing (LeDoux, 2014; LeDoux & Pine, 2016; Mobbs et al., 2019) as outlined in the introduction and method part. Thus, though we did not pre-register the study, we followed rigorous and principled procedures that eliminated analytic flexibility during the validation tests that established the decoder's accuracy. We attest that only the single, final model (i.e., VIFS) was tested on generalization datasets, and we did not try multiple models in order to find the best one (which would overfit the data). Thus, our analyses were conducted in a rigorous manner, and one that aligns with the aims that pre-registration is intended to achieve.

Based on the comment from the reviewer we included the following limitation in the revised manuscript as follows:

"Pre-registration has been increasingly advocated in the field of neuroimaging prediction studies (see e.g., also recent recommendations by Poldrack, Huckins, and Varoquaux (2020)) and might help to reduce analytic flexibility in neuroimaging analyses (Botvinik-Nezer et al., 2020). The analytic protocols for the present study have been established in our previous studies (see e.g., Chang et al., 2015; Wager et al., 2013; Zhou et al., 2020) and only the single final model was tested on the validation and generalization datasets. However, pre-registration in future studies could further facilitate analytic rigor."

The interpretation of weights of such decoding models is not so straightforward even using their Bootstrap procedure (see Kriegeskorte and Douglas, 2019, <https://www.sciencedirect.com/science/article/abs/pii/S0959438818301004>).

Essentially, high weights can be due to complex interactions with other features. As a result, most papers do not report or interpret weights.

Response: We agree with the reviewer that decoding models are subject to limitations in interpretation (Haufe et al., 2014; Kriegeskorte & Douglas, 2019), as are coefficients estimated in multiple regression more generally (Courville & Thompson, 2001). To (partly) account for these limitations in interpretability we employed convergent analyses, using both thresholded weights and model encoding maps to interpret model features, as advocated by recent recommendations on MVPA (e.g., Ashar, Andrews-Hanna, Dimidjian, & Wager, 2017; Haufe et al., 2014) and traditional statistical literature on interpreting multiple regression coefficients (Courville & Thompson, 2001).

Specifically, we supplemented bootstrap tests that identify voxels that make consistent contributions to prediction with group tests of within-participant predictions, identifying voxels that are predictive across participants, treating participant as a random effect. Of note, we did not propose that the significant features/voxels represent subjective fear-related important/necessary predictors. As the reviewer correctly pointed out, significant voxels do not necessarily reflect subjective fear but instead could, e.g., reach significance due to their role in suppressing noise in the data. To further improve the interpretability and specificity of the findings we employed a reconstruction procedure to convert the backward maps (i.e., prediction weights) to the forward maps (i.e., model encoding maps). Model encoding maps provide a univariate test identifying voxels that significantly encode model predictions, without controlling for other voxels/brain components (Haufe et al., 2014). They therefore complement bootstrap-thresholded model weights. To improve the interpretability of the findings we generally focus on overlapping regions between backward and transformed forward models, which theoretically represent regions that are reliably (bootstrap test) and consistently (one-sample t-test on within-subject models) predictive of (multivariate weight maps) and associated with (model encoding maps; see also traditional univariate analysis) subjective fear. For a further elaboration on the forward and backward approach see also response to the final comment of Reviewer #2.

Moreover, unlike the univariate analysis, the multivariate regression model can, to some extent, reveal whether a brain region is associated with the outcome while “controlling for the potential influence of other regions”. The intersection of multivariate weight maps and model encoding maps thus has the highest probability of reflecting regions (on the voxel level) which represent important contributors to a mental process. To facilitate the determination of the neural basis of subjective fear we therefore employed converging methods including univariate and multivariate models. The corresponding convergent findings from all analyses are displayed in Fig. 3 and supplementary Fig. 3 which serve to visualize the overlap between the identified brain patterns across analyses and represents the basis for our discussion. To acknowledge the limitations of the separate models and to emphasize our approach we included the following explanation in the revised version:

“To facilitate the determination and interpretation of a subjective fear signature convergent univariate and multivariate approaches were implemented. Spatial patterns (or regions)

that were consistently observed across backward and forward models were considered as reliably and consistently associated and predictive of subjective fear.”

The correlation between predicted and actual values is not the best metric to assess the predictive power of the signature. They should consider using the coefficient of determination (R2) (see Poldrack, Huckins, & Varoquaux, 2019, JAMA <https://jamanetwork.com/journals/jamapsychiatry/article-abstract/2756204>).

Response: We agree with the reviewer that the prediction-outcome correlation represents not the best metric to assess the prediction accuracy of the signature. Due to the limitations of the prediction-outcome correlation we actually reported root mean squared error (RMSE) and explained variance scores, which are very similar to R2 (see below for details) and have been implemented in several widely used approaches to evaluate model performance by machine learning software including e.g., scikit-learn (https://scikit-learn.org/stable/modules/model_evaluation.html#explained-variance-score).

$$R^2(\mathbf{y}, \hat{\mathbf{y}}) = 1 - \frac{\sum_{i=1}^n (\mathbf{y}_i - \hat{\mathbf{y}}_i)^2}{\sum_{i=1}^n (\mathbf{y}_i - \bar{\mathbf{y}})^2} = 1 - \frac{\frac{\sum_{i=1}^n (\mathbf{y}_i - \hat{\mathbf{y}}_i)^2}{n-1}}{\frac{\sum_{i=1}^n (\mathbf{y}_i - \bar{\mathbf{y}})^2}{n-1}} = 1 - \frac{\frac{\sum_{i=1}^n (\mathbf{y}_i - \hat{\mathbf{y}}_i)^2}{n-1}}{\text{var}(\mathbf{y})}$$

$$EVS(\mathbf{y}, \hat{\mathbf{y}}) = 1 - \frac{\text{var}(\mathbf{y} - \hat{\mathbf{y}})}{\text{var}(\mathbf{y})} = 1 - \frac{\frac{\sum_{i=1}^n (\mathbf{y}_i - \hat{\mathbf{y}}_i - \text{mean_error})^2}{n-1}}{\text{var}(\mathbf{y})}$$

Where EVS is the explained variance score, and var(y) indicates the variance of y. Thus, the EVS captures the out-of-sample variance explained (like prediction R²), and the RMSE is also recommended as a suitable error metric.

Of note, although several metrics and indices for assessing the predictive power of fMRI brain signatures have been discussed, there is currently no single gold standard and each metric comes with advantages and disadvantages. For instance, R2 can be considered as a combined measure of both the goodness of fit in the sense of how close the points fit the regression surface $\sum_{i=1}^n (\mathbf{y}_i - \hat{\mathbf{y}}_i)^2$, and the slope of the regression surface $\sum_{i=1}^n (\mathbf{y}_i - \bar{\mathbf{y}})^2$. Therefore, if the goodness of fit about the regression surface is fixed, a steeper surface $\sum_{i=1}^n (\mathbf{y}_i - \bar{\mathbf{y}})^2$ will increase and in turn increase R2 (Barrett, 1974).

Moreover, the application of the R2 (or the EVS) are limited due to properties of the fMRI signal, specifically scaling issues. For instance changes in the scale of the response across observations (due, e.g., to changes in the paradigm, scanner or even the 1st level GLM), the R2 (and EVS) will be poor due to inter-subject differences in scale. In this case the correlation might be useful as a index of within-person predictive accuracy (e.g., predicting the generalization dataset or the single-trial ratings in the current study). For this reason, we report this metric in addition to the others. In addition, as we are interested in within-subject subjective fear predictions we also reported forced-choice classification accuracies. To account for the advantages and disadvantages of the various prediction indices we employed several metrics (correlation, RMSE, EVS and classification accuracy) to facilitate

the evaluation of the predictive accuracy of the signature. To clarify this approach we included the following sentence in the methods:

“Several metrics have been proposed to evaluate the predictive power of multivariate predictive signatures (see e.g., Poldrack et al., 2020), however, the advantages and disadvantages of each metric are still a matter of debate, and metrics vary subtly in their properties. To facilitate a robust determination of the predictive accuracy of the neurofunctional signature we therefore employed various metrics including correlation, RMSE, EVS and forced-choice classification accuracy.”

The analyses presented in Figure 3 are not easy to interpret. They should present their combined map clearly and move the remaining maps (that were used to generate the combined one) to the supplementary material.

Response: We agree with the reviewer that the initial figure was not optimal. Our intention was to emphasize our approach described under response #1, that is to display the brain regions identified by different approaches as well as their convergence. We thank the reviewer for this advice and included an updated figure that only displays the key findings (VIFS, transformed model encoding and the conjunction maps) while presenting the further detailed results in supplementary figures (see supplementary Fig. 3).

Reviewer #2 (Remarks to the Author):

Review of Zhou et al., “A distributed fMRI-based neuromarker for the subjective experience of fear”

In this study, the authors develop and validate a model that can decode (across participants) the subjective experience of fear from fMRI data. The authors also distinguish the distributed neural pattern that corresponds to subjective fear from the neural representations of conditioned threat and negative affect, and they show that the output of their neural decoder mediates the association between the general negative affect response and subjective fear. Overall, I thought this was a very nice paper; it is closely in the spirit of Wager’s other work (e.g., identifying distributed neural correlates of pain ratings) and -- like that other work -- should have strong clinical utility. I don’t have any major concerns, but I have listed some miscellaneous points below (none of the suggested analyses are mandatory)

line 92: “V. T. decoder” -- sorry that I missed this, but what does V. T. stand for?

Response: We apologize for the unclear naming of the decoder. V.T. represent the name of the first author (Vincent Taschereau-Dumouchel) of a previous study that developed a fear predictive decoder (Taschereau-Dumouchel, Kawato, & Lau, 2020) and we used data from this study as generalization dataset and to benchmark our decoder. Briefly, the Taschereau-Dumouchel and colleagues developed a fear decoder based on pre-scanning ratings, which

captured not general but rather animal category-specific fear responses. In the present study we employed the decoder and the data from this study to test the generalizability of our decoder across different paradigms, scanners and cohorts. In line with the naming of other decoders in our manuscript we now included a naming and abbreviation which refers to the decoded context (animal fear schema signature, AFSS) in the revised manuscript.

line 98: "been challenged BY"

Response: We thank the reviewer for pointing out this mistake and corrected it in the revised version.

did the authors consider doing hyperparameter optimization (e.g., of the "C" parameter) through nested cross-validation?

Response: We thank the reviewer for this interesting suggestion. We did not include an optimization step in the initial version given that in our previous experience (1) results are often insensitive to the choice of C as long as C is in an intermediate range; and (2) optimization can introduce additional complexity, often without substantially improving accuracy. In addition, we wanted to eliminate researcher degrees of freedom/analysis flexibility where possible, so we elected to use the modal value of $C = 1$. Based on the comment from the reviewer we performed additional post hoc analyses optimizing the C parameter using nested cross-validation (across $C = [0.0001, 0.001, 0.01, 0.1, 1, 10]$). This procedure did not lead to strong improvements in the discovery cohort ($C = 1$ prediction-outcome correlation $r = 0.5647$; cross-validated $C r = 0.5648$). We additionally chose the optimal C parameter using the whole discovery cohort (evaluated by repeated 10-fold cross validation) and predicted the validation and generalization cohorts with the best model. We found that it predicted the validation cohort slightly better ($r = 0.5995$ vs. 0.5926 with $C = 1$), but predicted the generalization cohort slightly worse ($r = 0.5566$ vs. 0.5627 with $C = 1$). Given that our results remained stable across optimization methods we decided to include the new findings in the supplements in a specific section entitled 'Post hoc exploration of predictive models' for interested readers.

did the authors consider using recursive feature elimination to find the most compact set of informative voxels?

Response: Thanks for suggesting this additional interesting approach. RFE might indeed be helpful to identify the informative voxels. Based on the comment from the reviewer we performed a corresponding RFE analysis in which the importance of a feature was estimated by the absolute value of its corresponding predictive weight, and less important features (i.e., with low predictive weights) were eliminated recursively. We removed 10,000 voxels each time and re-trained the model with the remaining voxels and predicted validation and generalization cohorts until less than 10,000 voxels were left after the elimination (we further trained the model with the most important 10,000 voxels). We found that as the number of voxels/features decreased the prediction-outcome correlation of the validation cohort also decreased, but that predictive accuracy for the generalization cohort slightly increased until around 200,000 out of 225,672 voxels were removed (see figures below).

Moreover, in line with a previous study (Kohoutová et al., 2020) we plotted the final predictive SVR features after the RFE procedure (see figures below), with the final number of features = 20,000 (around 10% of the whole brain). The most compact set of informative voxels was very similar to the reliable voxels revealed by the bootstrap test, suggesting that the informative voxels were consistent across analytical approaches. However, given that (1) the number of most informative voxels is arbitrary and (2) the corresponding analysis was not introduced in the initial version of the manuscript (see also comment of reviewer #1 regarding pre-registration of prediction studies) we only include the new findings in the supplements of the revised version for interested authors.

(A) Model performance after the RFE procedure

(B) RFE weight map

Supplementary Fig. 5 Model performance and SVR features after the recursive feature elimination procedure. (A) Model performance after each elimination (10,000 voxels). Red line indicates the predictions on validation cohort and the blue line indicates the predictions on generalization cohort. (B) Weight map showing the final predictive SVR features after the RFE procedure, with the final number of features = 20,000.

another technique that might be useful for refining the pattern used by the classifier is SOS-Lasso, which imposes a prior that the neural pattern should be sparse but also locally structured; see <https://www.jneurosci.org/content/41/5/1019.abstract>

Response: Thank you for bringing the SOS-Lasso method to our attention. However, given the large number of alternative models that can be used for predictions (e.g., SOS-Lasso, Lasso-PCR, SVR, Elastic-Net, etc.) a comparison between the performance of the different models in the present and other datasets would be needed to guide the choice of model. However, this would be outside the scope of our study and, to further limit analytic flexibility (see also comment of reviewer #1 on preregistration) we did not include these further analyses in the revised version. However, we have cited this paper as an example of an interesting method that could be explored in future work as follows:

"...In the current study we used a SVR model to develop the VIFS and to explore the neural basis of subjective fear, however, the prediction accuracy and the contributing brain regions could be further explored by means of other candidate techniques such as SOS-LASSO which imposes a prior that the neural pattern should be sparse but also locally structured (Cox & Rogers, 2021).

lines 171-172: clarify that the predicted ratings were averaged as a function of reported fear rating within subjects before the within-subject correlations were computed (or, if I am misunderstanding, please clarify)

Response: We included one regressor for each unique rating (i.e., ratings ranging from 1-5) on the first level. For the within-subject correlations we calculated the correlation between the VIFS responses (i.e., predicted subjective fear) of five beta values (corresponding to conditions "rating 1", "rating 2", etc.) and corresponding self-report values (i.e., 1-5) for each subject separately. Thus, the predicted ratings were not averaged.

line 277: I would like to see a bit more info about the forward modeling technique in the main paper, as I was unfamiliar with the particular method used here

Response: We thank the reviewer for this comment and agree that further information should be provided in the main text. Generally speaking the GLM-based univariate analyses are forward models and the multivariate classification and regression techniques like SVM and SVR are backward models. Haufe et al. (2014) proposed a formula to transform linear backward models into linear forward models to facilitate the interpretation of the results in terms of neural mechanisms. The transformation employs the following formula: $A = cov(X) * W * cov(S)^{-1}$, where A is the reconstructed activation pattern (i.e., the transformed forward model), cov(X) is the covariance matrix of training data, W is the pattern weight vector, and cov(S) is the covariance matrix of the latent factors, which is defined as $W^T * X$. This is equivalent to assessing the relationship between each voxel and the fitted response in the multivariate model (i.e., overall model response). Specifically, for each voxel, one could use the beta values as Y and the predicted responses as X to run a GLM analysis for each voxel separately and the resulting beta value for the predicted responses would be the reconstructed activation for the corresponding voxel.

This reconstructed activation is also similar to the 'structure coefficients' in the statistical literature. Previous studies have argued that both betas and structure coefficients interpret the model (Courville & Thompson, 2001). Essentially, the beta tells you the predictive slope and direction of effect controlling for other variables in the model. The structure coefficients (model encoding value) tells you about the direction of the relationship between the variable and the model without controlling for other variables – i.e., which voxels are positively and which are negatively related to the predicted value.

line 774: "in parallel with within-subject models"

line 1062: "Panel B SUMMARIZES"

Response: we apologize for the typos and corrected them in the revised manuscript.

Sincerely,

Ken Norman (I sign all of my reviews)

Reviewer #3 (Remarks to the Author):

In this study, BOLD-fMRI data from multiple cohorts of subjects and protocols (and indeed scanners) were leveraged to argue for a generalizable multivoxel signature of visually induced fear. The first study used a validated set of visual stimuli that evoke parametric ratings of subjective fear experience. In 67 subjects this was used to discover the multivoxel features that could predict fear, using a linear predictive model; in addition to the cross-validated initial findings, an independent confirmation cohort (N=20) was then also used (decoder fully trained on discovery sample, tested on this independently held out sample of 20). The predictive accuracy was very high (e.g., $r > 0.8$) and dichotomizing the fear ratings to high vs. low yielded classification performance of 100%. The VIFS did not depend on occipital cortex and emerged about 4sec after image onset; it generalized to predicting fear in other datasets, even acquired on other scanners. Finally, this population-trained

VIFS was applied to the trial wise beta maps in each individual subjects, and was able to predict, on each trial, the intensity of fear experience for most subjects (r s around 0.4). Overall, this suggested a sensitive and generalizable set of neural features that a linear model could use to predict subjective fear in general and individually.

The authors next investigated the neuroanatomical substrates of their findings. Standard univariate analyses showed activations and deactivations to fear distributed over many structures, as expected. These mostly overlapped with the maps obtained from the weights of the MVPA features obtained from the above predictive models. To control for the possibility that these feature weights may not selectively reflect fear intensity (but may be predictive because they cancel out noise from other voxels, a well-known issue in interpreting the weights of regularized regressions), they also computed the structure coefficients ("forward model"). Bottom line: these all looked very similar.

Another set of approaches used searchlights or ROIs. Even when carefully comparing for the number of features, the VIFS always performed a lot better than any of these more anatomically circumscribed sets of features (although they could predict to some extent). It was found that about 10,000 voxels reached optimal predictive accuracy, provided they were sampled all over the brain.

Overall, I find this a technically impressive study. However, I have questions about the interpretation, as well as some others:

1. I wonder about the validity and specificity of the stimuli and ratings. IAPS-type images were used, which is not ideal (videos are much better). As far as I can tell, subjects were not given a choice or freely asked how they felt when they looked at these, but forced only to give a rating on fear. So how do we know how specific this is to fear? It would seem that the findings could also be just explained by arousal — could the authors please address this question. The construct of fear is not well assessed with these stimuli or questions, since no other emotion dimensions seem to have been assessed. This stands in contrast to the detailed neuroanatomical investigation to show how specific (or distributed) anatomical features are— but no equivalent dissection is undertaken for “fear” leaving the validity and specificity to fear rather understudied here. The study shows that the VIFS is predicting something, but it is far from clear that what it is predicting is specific to fear, as far as I can tell.

Response: We thank the reviewer for raising important points regarding the stimuli and we fully agree that it is necessary to further demonstrate the specificity of the neurofunctional decoder with respect to fear. We generally agree with the reviewer that employing dynamic (video) stimuli would have led to a stronger induction of emotions. However, we chose static stimuli for designing the present decoder to increase comparability with already established decoders for threat conditioning (Reddan, Wager, & Schiller, 2018) and general negative affect (Chang et al., 2015) and thus allow to examine separable neurofunctional representations of these processes. These decoders have been previously developed based on static (pictorial) stimuli and comparing these decoders with a subjective fear decoder based on dynamic stimuli such as movies would have introduced an additional source of variance of unknown impact on the comparison of the decoders. Nevertheless we agree with the reviewer that dynamic stimuli may have the ability to induce stronger emotional experience in the participants. To this end we included the following limitation in the revised manuscript:

“...The present study used IAPS-type static images as stimuli. Although ratings revealed that these images could elicit a relative robust range of subjective fear experience, the types of variations in stimuli that lead to distinct vs. similar neural encoding are still not well understood. It is for instance conceivable that video stimuli could activate the VIFS in proportion to the fear-inducing properties of the videos, or it is possible that the brain encodes dynamic stimuli differently. These possibilities could be tested in future studies.”

Secondly, we agree with the reviewer that the IAPS-type images may have elicit other negative feelings like e.g. disgust or nonspecific emotional arousal in the participants and that this in turn would have influenced the decoder. Based on the comment from the reviewer, we acquired additional data to better understand to what extent the VIFS relates or reacts to other emotional experiences. To this end we acquired ratings of associated negative emotions for the stimuli in an independent sample of participants (n = 120). Participants were required to rate the level of subjective disgust, anger, sadness, arousal, valence and fear induced by the pictures. The fear ratings were included to further test replication and robustness across samples. The ratings were acquired online and each participant rated all stimuli with respect to one emotion (n = 20 subjects per emotion). Ratings were provided on a 5-point rating scale ranging from “1” (not at all) to “5”

(extremely) for all dimensions except for valence which was rated from "1" (extremely positive) to "9" (extremely negative) with "5" indicating neutral. To determine whether and to which extent the VIFS reacts to other emotional domains, we correlated the image-by-image series of normative ratings with the image-by-image variation in VIFS responses, for each emotion category assessed. Specifically, we used the single trial beta maps for each picture and averaged the cross-validated VIFS responses for each picture (i.e., we used a cross-validation procedure, where we trained the VIFS with data from ~90% subjects and applied this model to calculate the VIFS response for the remaining subjects). We next correlated the picture-specific group-average VIFS responses with the picture-specific group-average ratings for each emotional domain separately (for a similar approach see Ashar et al., 2017). The VIFS response was more strongly correlated with subjective fear ($r_{79} = 0.77$) than any other emotional rating (disgust: $r_{79} = 0.64$; anger: $r_{79} = 0.63$; sadness: $r_{79} = 0.60$; arousal: $r_{79} = 0.66$; valence: $r_{79} = 0.65$) suggesting that the VIFS indeed reacts most strongly to subjective fear and to a lesser extent to other related negative emotions or general emotional features such as arousal.

In addition to the correlations between group-average trial-by-trial VIFS and ratings we calculated the correlations within each subject and direct comparisons of the correlations between VIFS and emotion ratings supported our conclusion and revealed significantly stronger correlations with fear than other emotions. Specifically, For each subject in the discovery cohort, we correlated the cross-validated VIFS response for each picture with the picture-specific group-average ratings for each emotional domain separately. We found that the VIFS tracked subjective fear ratings significantly better than any of the other 5 emotions collected in the online sample (e.g. fear versus the second best prediction, arousal; paired t-test $t_{66} = 7.31$, $P < 0.001$, Bonferroni corrected). Together with our previous findings showing that (1) the VIFS could not distinguish CS+ (which induces higher autonomic responses as reflected in elevated SCR responses) from CS- and (2) the prediction accuracy of VIFS on high arousing nonspecific negative emotion ($r = 0.29$) was substantially lower than the prediction accuracies of the subjective fear ($r > 0.55$), these findings suggest that the VIFS shows reasonable specificity for subjective fear, but to some extent also captures aspects of other negative emotions or arousal which are inherently associated with fear.

We included the corresponding new results and a brief discussion on the specificity of the decoder for fear as follows in the revised version of the manuscript:

Specificity of the VIFS for the experience of fear

Given that emotional stimuli such as the pictures we used can induce a complex array of negative emotional experiences (e.g. disgust, anger, nonspecific negative arousal) we further explored whether the VIFS is most closely related to subjective fear. To this end we acquired ratings of several other negative emotions (disgust, anger and sadness), valence, and arousal in an independent sample of participants ($n = 120$). The ratings were acquired online and each participant rated all stimuli with respect to one emotion ($n = 20$ participants per emotion). Ratings were provided on a 5-point rating scale ranging from "1" (not at all)

to "5" (extremely) for all dimensions except for valence which was rated from "1" (extremely positive) to "9" (extremely negative) with "5" indicating neutral. To determine whether and to which extent the VIFS reacts to other emotional domains, we correlated the image-by-image series of normative ratings with the image-by-image variation in VIFS responses, for each emotion category assessed. Specifically, we used the single trial beta maps for each picture and averaged the cross-validated VIFS responses for each picture. We next correlated the picture-specific group-average VIFS responses with the picture-specific group-average ratings for each emotional domain separately (for a similar approach see Ashar et al., 2017). The VIFS response was more strongly correlated with subjective fear ($r_{79} = 0.77$) than any other emotional rating (disgust: $r_{79} = 0.64$; anger: $r_{79} = 0.63$; sadness: $r_{79} = 0.60$; arousal: $r_{79} = 0.66$; valence: $r_{79} = 0.65$) suggesting that the VIFS indeed reacts most strongly to subjective fear and to a lesser extent to other related negative emotions or general emotional features such as arousal. Moreover, Direct comparisons of the correlations between VIFS and emotion ratings supported this conclusion and revealed significantly stronger correlations with fear than other emotions. For each subject in the discovery cohort, we correlated the cross-validated VIFS response for each picture with the picture-specific group-average ratings for each emotional domain separately. We found that the VIFS tracked subjective fear ratings significantly better than any of the other 5 emotions collected in the online sample (e.g. fear versus the second best prediction, arousal; paired t-test $t_{66} = 7.31$, $P < 0.001$, Bonferroni corrected). Together with our previous findings showing that (1) the VIFS could not distinguish CS+ (which induces higher autonomic responses as reflected in elevated SCR responses) from CS- and (2) the prediction accuracy of VIFS on high arousing nonspecific negative emotion was substantially lower than the prediction accuracies of the subjective fear, these findings suggest that the VIFS shows reasonable specificity for subjective fear, but to some extent also captures aspects of other negative emotions or arousal which are inherently associated with fear."

2. The comparison to the conditioning study is a little hard to interpret, since different objectives are used. For the present fear study, it is the parametric rating of experienced fear, irrespective of the stimuli. For the conditioning study, it was the CS+ or CS- without any fear ratings. This strikes me like comparing apples and oranges. What would be a better comparison would be to compare to actual ratings of subjective threat experience in the conditioning study.

Response: We agree with the reviewer that different objectives and correspondingly different paradigms were used in the subjective fear and fear conditioning studies. Although these paradigms serve different objectives both paradigms have been used widely in the literature to image fear networks and engage overlapping neural systems associated with fear such as the amygdala, insula and prefrontal regions (Fullana et al., 2016; Reddan et al., 2018). Aligning the paradigms in terms of their trial structure and ratings would blur the distinction between non-conscious threat-related processing and conscious appraisals. For instance, the inclusion of subjective fear ratings following each trial during the conditioning paradigm is not commonly employed in conditioning procedures which usually use autonomic threat responses as measured by SCR as key behavioral index. In fact, the inclusion of explicit fear/threat ratings could bias the conditioning procedure – away from a

hardwired defensive response towards a process of subjective fear experience and evaluation, which would be against our key question of whether the processes and neural bases of the subjective experience of fear and the automatic (hard-wired) threat response during conditioning are distinct (LeDoux, 2014; LeDoux & Pine, 2016; Mobbs et al., 2019).

To partially overcome the difficulty that the paradigms and stimuli for the different fear facets are inherently different we developed a novel cross-prediction approach. In the current study we showed that although the VIFS was developed on continuous fear ratings it could still be used to predict categorical fearful pictures in the generalization cohort (high vs low fear induction in the absence of subjective fear ratings). The structure of this paradigm, particularly the presentation of categorical stimuli in the absence of ratings, thus corresponds closely to the fear conditioning paradigms. In one of our previous studies we moreover demonstrated that a decoder developed on categorical stimuli (vicarious pain vs. control) could predict continuous emotional ratings (self-experienced thermal pain; prediction-outcome correlation $r = 0.54$) (Zhou et al., 2020), suggesting that a prediction across different objectives and paradigms is generally feasible. Together, our findings suggest that a cross-prediction could work if mental processes share common neural representations in despite of different paradigms or stimuli. Therefore we hypothesized that if subjective fear and conditioned threat shared common neural representations the VIFS should significantly predict conditioned threat and the TPS (threat predictive signature) should predict the levels of subjective fear. Conversely, low cross-prediction indicates independence of the neural representations for 'fear' and 'conditioned threat' constructs (for similar approaches see e.g., Corradi-Dell'Acqua, Tusche, Vuilleumier, & Singer, 2016; Krishnan et al., 2016; Woo et al., 2014).

We acknowledge that this represents a more indirect approach and one potential solution of minimizing the differences between paradigms could have been to employ categorical stimuli without ratings for both paradigms (e.g., high subjective fear vs. neutral stimuli with post-fMRI ratings for the fear experience paradigm). However, we deliberately decided to employ a regression model to develop the VIFS in the current study because this allowed us to increase the specificity for our primary decoder for subjective fear given that multivariate classification models based on categorical stimuli tend to reflect a range of processes that differ between the categorical conditions and are more likely to capture activity related to arousal and attention (Kragel et al., 2018).

Based on the comment from the reviewer we included the following limitation in the revised version:

"... although we identified distinct neural representations for subjective fear and conditioned threat on the whole-brain level the corresponding decoders were developed based on studies employing different paradigms and stimuli. The independence of common neurofunctional representations of subjective fear and conditioned threat thus needs to be further evaluated. Future studies could, e.g., align the paradigms by using categorical stimuli across the paradigms (e.g., high fear vs. neutral stimuli) to further explore whether subjective fear and conditioned threat share common neural representations, particularly in local regions. However, the specificity of the shared neural basis (if one is found) to threat- and fear-related processes of interest would require further testing."

3. Both for the comparison to the conditioning study, and for the comparison to the negative affect study, these comparisons are not ideal. In each case, a predictive model was optimized to each dataset alone, and then the features or cross-decoding were compared. But to answer the question of whether or not there are shared representations, a joint model should be trained — that is, we should ask, if we want to predict BOTH fear and negative affect, can we train such a model and how well does it do. Such a jointly trained model could well draw on features that are quite different from those in the models based on any one dataset in isolation.

Response: We agree with the reviewer that a jointly trained model could identify shared neural representations, however, this model may be limited with respect to specificity because the overarching decoder will likely capture shared nonspecific processes like arousal. We employed a corresponding joint-model approach in one of our previous studies to determine common neural representations of pain empathy across different pain empathy inducing stimuli and next further tested the specificity against nonspecific negative affect decoder (Zhou et al., 2020). However, in the present study we decided to employ a cross-dataset prediction approach because the technical challenges permit the development of a joint model using the current datasets. For instance, the scaling issue of the fMRI signal between different scanners will likely induce variance between the datasets unrelated to the specific mental process and affect the prediction performance. Nevertheless, we agree with the reviewer that the development of overarching decoders represents an interesting direction for future studies to identify shared neural basis for fear and (non-fearful) negative affect and include this as future direction in the revised manuscript (see below).

As for the comparison between subjective fear and conditioned threat (see also response to comment #2), in addition to the scaling issue of the fMRI signal, in the current study the CS+/- classification is categorical while the fear rating prediction is based on continuous fear ratings and from a technical point of view the integration of these into a single model may also be problematic. Future studies could employ e.g., a joint classification model to predict CS+ and high fear from CS- and low fear. However, as compared to regression models (e.g., SVR in the current study) such classification models tend to reflect a range of processes that differ between the conditions and are more likely to capture activity related to arousal and attention (Kragel et al., 2018) and therefore the specificity of the shared neural basis (if one is found) to threat- and fear-related processes of interest would require further testing.

We included a future direction in the revised manuscript as follows (the limitation of the comparison between subjective fear and conditioned threat please see last response):

“...In the current study we showed that subjective fear and nonspecific negative emotion shared common yet also distinct neural representations. Our findings are based on cross-prediction models and training joint-models over the emotional domains in datasets that have been acquired with matched paradigms and on an identical MRI system may help to further determine common and separable neural representations between fear experience and other emotional domains.”

4. The analyses are sophisticated and many checks have been put in place to ensure reliability. But the question of validity remains: what exactly does the VIFS predict? The authors note some of the debates in the field at the beginning of their paper. The fact that their results generalize, but are distinct from conditioned threat or negative affect, all argue that whatever is being predicted, it is something that accompanies fear. But what exactly is that? As is common, the authors refer to the VIFS as a “neural representation” of fear, but this seems far from clear, especially given the debates. I am also somewhat puzzled by the rather long latency to find the VIFS (ca. 4 seconds after stimulus onset), which would be more consistent with autonomic or motoric consequences of fear rather than fear itself. Finally, I find the extremely high predictive accuracy of the VIFS puzzling. Surely not everybody is experiencing fear the same way from these stimuli?

Response: The 4s latency of the VIFS response is expected given that the BOLD response rises to a peak over 4–6s. We employed forced-choice classification, where the VIFS responses were compared for two conditions tested within the same individual, and the higher one was chosen as more fearful, to calculate the predictive accuracy. The high accuracies were consistent with the studies using the same method (Chang et al., 2015; Wager et al., 2013). The high forced-choice accuracy actually indicates that the VIFS response increases as a function of increasing subjective fear on the individual level. We agree with the reviewer that large individual variations exist with respect to the induction of subjective fear by specific stimuli, including the pictures that were employed in the present study. Accordingly, we did not include pre-specified or group level ratings in the analysis but rather the individual subjective fear ratings were employed throughout all (univariate as well as multivariate) analyses. Thus, which images are “high fear” varies across individuals, so individuals may vary substantially in which stimuli induce fear without reducing the decoder’s accuracy. With respect to the specificity of the neurofunctional representations please refer to the discussion under response #1 and #2. To clarify the latency of the response we include the following information in the legend of Fig. 2.

Figure 2D and 2E, please note that the VIFS reacts with a latency of approximate 4 seconds after stimulus onset which corresponds to the timing of the hemodynamic response function (HRF) following stimulus onset.

5. The discussion and conclusions of the paper should include a more nuanced discussion of how these findings relate to debates in the literature, and of the limitation of the present findings. A lot of the literature on fear is based on data from rodents, and it is certainly possible that the rodent brain “represents” fear differently than does the human brain. The present findings are all based on fMRI, which is indirect, has poor spatiotemporal resolution, and, most important, shows only correlations. So it could be quite possible that the amygdala is causally necessary for fear, even subjective fear in humans, even though many other brain regions carry information about subjective fear once it has been caused.

Response: We agree with the reviewer that the conclusions and limitations of the current study should be more clearly discussed in the manuscript and that the low spatiotemporal resolution of fMRI may not allow us to clearly separate regions representing initial fear

detection modules from regions which are subsequently engaged. Based on the comments from the reviewer, we further elaborated on the potential limitations of the present study within the context of human amygdala lesion studies (which circumvent the problem of spatial resolution because in patients with focal and complete amygdala lesions the amygdala as fear response trigger is absent) and ongoing debates on the critical role of the amygdala in fear, specifically between LeDoux (e.g., LeDoux & Pine, 2016) and Fanselow (e.g., Fanselow & Pennington, 2017). The corresponding discussion has been incorporated as follows:

"...The amygdala is often considered to be a 'fear center' or 'threat center' in animal models (for a critical discussion on the role of the amygdala in fear and threat see also (LeDoux & Pine, 2016). Although a direct translation of threat-related neural representations in rodents to human emotional experiences is limited, a number of human lesion studies in patients with complete bilateral amygdala lesions underscores the complex role of the amygdala in fear processing in humans. In line with the 'fear center' perspective, an early human lesion study showed that a patient with focal bilateral amygdala lesions never endorsed feeling more than minimal levels of fear (Justin S. Feinstein, Adolphs, Damasio, & Tranel, 2011). However, other studies in patients with bilateral focal and complete amygdala lesions demonstrated that the amygdala was not critically required to experience panic triggered by a CO₂ challenge (J. S. Feinstein et al., 2013), subjective affective experience (Anderson & Phelps, 2002) or the modulation of the acoustic startle reflex by fear-inducing background stimuli (Becker et al., 2012), which together raise the question of whether the amygdala is causally necessary and sufficient for the experience of subjective fear in humans (for an in depth discussion see also (LeDoux & Pine, 2016)). Whereas our findings indicate that the amygdala per se is not sufficient to represent subjective experience of fear in humans, the question of a causal role of the amygdala in subjective fear in humans cannot be ultimately addressed in the present study given the indirect nature of fMRI measurements and lack of direct experimental manipulations of the brain. In addition to a widely distributed pattern of activity, voxels in the amygdala were identified across our analytic approaches, suggesting that the amygdala may represent a part of a larger network for initiating or integrating a coordinated fear and threat response on different levels (see e.g., Fanselow & Pennington, 2017). "

6. The actual protocol seems somewhat strange. In the discovery study, only 20 stimuli are used (in 4 runs), shown each for 6 seconds with a rating subsequently. That's a total of about 16 minutes of scanning. I would have expected considerably more data per subject, and a larger variety of stimuli. Also, I could not find any description of the low-level visual properties of the stimuli, which needs to be controlled for. Finally, asking subjects to give explicit ratings partly confounds conceptualization required in the task with the subjective experience— it would be important to compare these data to ones where no ratings are required during the scan (but subjects rate the stimuli again outside the scanner subsequently).

Response: We believe that there is a misunderstanding regarding the number of stimuli in the discovery cohort. We actually included 80 stimuli for the discovery cohort, 20 for each

run (please see Fig. 1A; we also reformulated the method part to clarify the number of stimuli). We thank the reviewer for pointing our attention to the potential influence of visual features on the prediction. Based on this comment we determined several visual features of the stimuli and tested whether these can be accurately predicted. In detail, we measured the edge intensity (MATLAB's Canny edge detector), the saliency (<http://www.saliencytoolbox.net/>) as well as the visual clutter (feature congestion and subband entropy (Rosenholtz, Li, & Nakano, 2007)) for each picture. Next, we ran correlational analyses as introduced in our response to the first comment of Reviewer #3. We found that the group-average VIFS responses were not significantly correlated with any of the visual features (most significant $r = -0.19$, $P = 0.09$). Moreover, the VIFS tracked ratings of subjective fear from the online sample significantly stronger than it tracked any of the visual features (fear versus the next closest feature, edge intensity; paired t-test $t_{66} = 22.15$, $P < 0.001$, Bonferroni corrected). Taken together, our findings suggest that the prediction performance was not driven by the visual properties of the stimuli. The new findings have been included in the "Specificity of the VIFS for the experience of fear" section.

Finally, we agree with the reviewer that asking subjects to give explicit ratings might partly limit the decoder to situations which require an explicit evaluation of the level of subjective fear. However, the inclusion of the generalization dataset and corresponding findings argue against this limitation. In the generalization dataset the authors briefly asked subjects outside the scanner "how much are you afraid of snakes, spiders, etc.", without showing the photos of the animals, and participants passively viewed images during scanning. Our decoder predicted the ratings of each animal category with similar accuracy as compared to the prediction using the generalization dataset itself (cross-validated), providing evidence that that the VIFS predicts the level of fear across both active rating and passive viewing conditions.

References

- Anderson, A. K., & Phelps, E. A. (2002). Is the human amygdala critical for the subjective experience of emotion? Evidence of intact dispositional affect in patients with amygdala lesions. *J Cogn Neurosci*, *14*(5), 709-720. doi: 10.1162/08989290260138618
- Ashar, Y. K., Andrews-Hanna, J. R., Dimidjian, S., & Wager, T. D. (2017). Empathic care and distress: predictive brain markers and dissociable brain systems. *Neuron*, *94*(6), 1263-1273. e1264.
- Barrett, J. P. (1974). The Coefficient of Determination—Some Limitations. *The American Statistician*, *28*(1), 19-20. doi: 10.1080/00031305.1974.10479056
- Becker, B., Mihov, Y., Scheele, D., Kendrick, K. M., Feinstein, J. S., Matusch, A., . . . Hurlmann, R. (2012). Fear processing and social networking in the absence of a functional amygdala. *Biol Psychiatry*, *72*(1), 70-77. doi: 10.1016/j.biopsych.2011.11.024
- Botvinik-Nezer, R., Holzmeister, F., Camerer, C. F., Dreber, A., Huber, J., Johannesson, M., . . . Schonberg, T. (2020). Variability in the analysis of a single neuroimaging dataset by many teams. *Nature*, *582*(7810), 84-88. doi: 10.1038/s41586-020-2314-9

- Chang, L. J., Gianaros, P. J., Manuck, S. B., Krishnan, A., & Wager, T. D. (2015). A sensitive and specific neural signature for picture-induced negative affect. *PLoS biology*, *13*(6), e1002180.
- Corradi-Dell'Acqua, C., Tusche, A., Vuilleumier, P., & Singer, T. (2016). Cross-modal representations of first-hand and vicarious pain, disgust and fairness in insular and cingulate cortex. *Nature communications*, *7*, 10904.
- Courville, T., & Thompson, B. (2001). Use of structure coefficients in published multiple regression articles: β is not enough. *Educational and Psychological Measurement*, *61*(2), 229-248.
- Cox, C. R., & Rogers, T. T. (2021). Finding Distributed Needles in Neural Haystacks. *The Journal of Neuroscience*, *41*(5), 1019-1032. doi: 10.1523/jneurosci.0904-20.2020
- Fanselow, M. S., & Pennington, Z. T. (2017). The Danger of LeDoux and Pine's Two-System Framework for Fear. *Am J Psychiatry*, *174*(11), 1120-1121. doi: 10.1176/appi.ajp.2017.17070818
- Feinstein, J. S., Adolphs, R., Damasio, A., & Tranel, D. (2011). The Human Amygdala and the Induction and Experience of Fear. *Current Biology*, *21*(1), 34-38. doi: <https://doi.org/10.1016/j.cub.2010.11.042>
- Feinstein, J. S., Buzza, C., Hurlemann, R., Follmer, R. L., Dahdaleh, N. S., Coryell, W. H., . . . Wemmie, J. A. (2013). Fear and panic in humans with bilateral amygdala damage. *Nat Neurosci*, *16*(3), 270-272. doi: 10.1038/nn.3323
- Fullana, M., Harrison, B., Soriano-Mas, C., Vervliet, B., Cardoner, N., Àvila-Parcet, A., & Radua, J. (2016). Neural signatures of human fear conditioning: an updated and extended meta-analysis of fMRI studies. *Molecular psychiatry*, *21*(4), 500-508.
- Haufe, S., Meinecke, F., Görgen, K., Dähne, S., Haynes, J.-D., Blankertz, B., & Bießmann, F. (2014). On the interpretation of weight vectors of linear models in multivariate neuroimaging. *Neuroimage*, *87*, 96-110.
- Kohoutová, L., Heo, J., Cha, S., Lee, S., Moon, T., Wager, T. D., & Woo, C.-W. (2020). Toward a unified framework for interpreting machine-learning models in neuroimaging. *Nature protocols*, *15*(4), 1399-1435. doi: 10.1038/s41596-019-0289-5
- Kragel, P. A., Kano, M., Van Oudenhove, L., Ly, H. G., Dupont, P., Rubio, A., . . . Gianaros, P. J. (2018). Generalizable representations of pain, cognitive control, and negative emotion in medial frontal cortex. *Nat Neurosci*, *1*.
- Kriegeskorte, N., & Douglas, P. K. (2019). Interpreting encoding and decoding models. *Curr Opin Neurobiol*, *55*, 167-179. doi: <https://doi.org/10.1016/j.conb.2019.04.002>
- Krishnan, A., Woo, C.-W., Chang, L. J., Ruzic, L., Gu, X., Lopez-Sola, M., . . . Wager, T. D. (2016). Somatic and vicarious pain are represented by dissociable multivariate brain patterns. *Elife*, *5*, e15166.
- LeDoux, J. E. (2014). Coming to terms with fear. *Proceedings of the National Academy of Sciences*, *111*(8), 2871-2878.
- LeDoux, J. E., & Pine, D. S. (2016). Using neuroscience to help understand fear and anxiety: a two-system framework. *American Journal of Psychiatry*.
- Mobbs, D., Adolphs, R., Fanselow, M. S., Barrett, L. F., LeDoux, J. E., Ressler, K., & Tye, K. M. (2019). Viewpoints: Approaches to defining and investigating fear. *Nat Neurosci*, *22*(8), 1205-1216.

- Poldrack, R. A., Huckins, G., & Varoquaux, G. (2020). Establishment of Best Practices for Evidence for Prediction: A Review. *JAMA psychiatry*, *77*(5), 534-540. doi: 10.1001/jamapsychiatry.2019.3671
- Reddan, M. C., Wager, T. D., & Schiller, D. (2018). Attenuating neural threat expression with imagination. *Neuron*, *100*(4), 994-1005. e1004.
- Rosenholtz, R., Li, Y., & Nakano, L. (2007). Measuring visual clutter. *J Vis*, *7*(2), 17.11-22. doi: 10.1167/7.2.17
- Taschereau-Dumouchel, V., Kawato, M., & Lau, H. (2020). Multivoxel pattern analysis reveals dissociations between subjective fear and its physiological correlates. *Molecular psychiatry*, *25*(10), 2342-2354.
- Wager, T. D., Atlas, L. Y., Lindquist, M. A., Roy, M., Woo, C.-W., & Kross, E. (2013). An fMRI-based neurologic signature of physical pain. *New England Journal of Medicine*, *368*(15), 1388-1397.
- Woo, C.-W., Koban, L., Kross, E., Lindquist, M. A., Banich, M. T., Ruzic, L., . . . Wager, T. D. (2014). Separate neural representations for physical pain and social rejection. *Nature communications*, *5*, 5380.
- Zhou, F., Li, J., Zhao, W., Xu, L., Zheng, X., Fu, M., . . . Becker, B. (2020). Empathic pain evoked by sensory and emotional-communicative cues share common and process-specific neural representations. *Elife*, *9*, e56929.

REVIEWER COMMENTS

Reviewer #1 (Remarks to the Author):

The authors addressed some of the methodological issues I raised (e.g., they now use primarily predictions from the external validation samples). But their argument regarding the superiority of correlation coefficients is somewhat inconsistent with what they did in the paper (i.e., they mostly used between-subject correlations and said it was best for within-subject correlations). Also, they mostly ignored our comments about providing a better discussion of the relation of their decoding findings with published work by others. If they can address these issues, I would support publication.

Reviewer #2 (Remarks to the Author):

The authors have satisfactorily addressed my comments from the previous round of reviews.
Sincerely,
Ken Norman (I sign all of my reviews)

Reviewer #3 (Remarks to the Author):

The authors have thoroughly revised their paper, and I have read both the revised paper and their rebuttal. The very detailed response to Reviewer #1 is especially impressive; the study was already impressive in its generalizability across samples to begin with. The authors have answered reasonably all of the questions that I had. I feel that this will be an important contribution to the literature and I have no further suggestions for improvement.

Reviewer #1 (Remarks to the Author):

The authors addressed some of the methodological issues I raised (e.g., they now use primarily predictions from the external validation samples). But their argument regarding the superiority of correlation coefficients is somewhat inconsistent with what they did in the paper (i.e., they mostly used between-subject correlations and said it was best for within-subject correlations). Also, they mostly ignored our comments about providing a better discussion of the relation of their decoding findings with published work by others. If they can address these issues, I would support publication.

Response: We thank the reviewer for these additional comments and agree that (a) from a methodological point more details and a more consistent approach, as well as (b) from a conceptual point a better integration and discussion of the findings in the context of the previous decoding literature will improve the manuscript.

(a) We employed RMSE, EVS, (overall and within-subject) prediction-outcome correlations and forced-choice classification accuracy to evaluate the performance of the VIFS on discovery and validation datasets which were acquired using the same MRI system and very similar paradigms. Of note, we used overall (that is within- and between-subject; i.e., the number of subjects * the number of conditions pairs; e.g., 333 pairs in the discovery cohort) as well as within-subject correlations, yet not between-subject correlations per se. Given that the scaling issues (e.g. due to different paradigms or MRI systems) may limit the application of the EVS or RMSE we however employed correlation and forced-choice classification approaches to test the generalization ability of the VIFS to another dataset acquired using a different MRI system and fear-induction paradigm (i.e., the generalization dataset). Based on the comment from the reviewer we realized that we only reported the overall correlation between VIFS' predictions and the subjective ratings in the generalization analyses but not the within-subject correlations. In the revised version we therefore include the within-subject prediction-outcome correlation coefficients of the VIFS in the generalization ($r = 0.65 \pm 0.06$) cohort. For a better comparison we also report within-subject prediction-outcome correlation coefficients for the AFSS (animal fear schema signature) in the discovery and validation datasets ($r = 0.41 \pm 0.05$ and $r = 0.60 \pm 0.06$, respectively) in the revised manuscript (see Supplementary Results for details). In addition, we included within-subject correlations (mean \pm SE) for the comparison between VIFS and PINES (Table 2).

As for the searchlight-, parcellation- and network-based analyses we only reported overall prediction-outcome correlations because the main purpose of these analyses was to show that distributed brain regions were predictive of subjective fear and that local predictions were substantially worse than the prediction using VIFS, and in this case comparing the single

effect size (i.e., overall correlation coefficient) of the VIFS with the effect size of each local model is more intuitive and informative. This approach is in line with previous studies showing that e.g., negative affect and pain are encoded in multiple brain systems (Chang, Gianaros, Manuck, Krishnan, & Wager, 2015; Kragel, Koban, Barrett, & Wager, 2018).

(b) We agree with the reviewer that the findings should be discussed in the context of the previous published work and extended the discussion as follows:

“A previous study (Taschereau-Dumouchel, Kawato, & Lau, 2019), which aimed at comparing the neural representations of subjective fear and the physiological threat response, developed a decoder (i.e., AFSS) predictive of reported fear as assessed by offline ratings to different animal categories. The VIFS generalized well to the dataset used to train the AFSS, but the AFSS did not generalize to the same extent to the datasets used to train the VIFS. This might be due to the fact that the AFSS reflects more “stable” fear schemas (e.g. the general fear of spiders) while the VIFS is more accurate to capture fear across different stimulus classes and contexts. Nevertheless, both studies have consistently demonstrated that activation patterns in distributed brain systems including e.g., prefrontal regions, insula and hippocampus are predictive of subjective fear and that activation in the ‘fear center’ (i.e., amygdala) is not sufficient to represent subjective fear experience. These findings are consistent with a growing number of studies demonstrating that brain activation in isolated regions or specific subsystem is neither necessary nor sufficient for predicting subjective emotional experiences (see e.g. also findings from a “virtual lesion” approach in (Chang et al., 2015)). Moreover, in line with our findings showing that subjective fear and conditioned threat responses engage distinct neural representations in humans Taschereau-Dumouchel et al. (2019) reported distinguishable neural representations of subjective fear and its physiological correlates, together suggesting separable neural representations of subjective fear experience and hard-wired defensive responses.”.

Reviewer #2 (Remarks to the Author):

The authors have satisfactorily addressed my comments from the previous round of reviews.

Sincerely,

Ken Norman (I sign all of my reviews)

Reviewer #3 (Remarks to the Author):

The authors have thoroughly revised their paper, and I have read both the revised paper and their rebuttal. The very detailed response to Reviewer #1 is especially impressive; the study

was already impressive in its generalizability across samples to begin with. The authors have answered reasonably all of the questions that I had. I feel that this will be an important contribution to the literature and I have no further suggestions for improvement.

References

- Chang, L. J., Gianaros, P. J., Manuck, S. B., Krishnan, A., & Wager, T. D. (2015). A sensitive and specific neural signature for picture-induced negative affect. *PLoS biology*, *13*(6), e1002180.
- Kragel, P. A., Koban, L., Barrett, L. F., & Wager, T. D. (2018). Representation, pattern information, and brain signatures: from neurons to neuroimaging. *Neuron*, *99*(2), 257-273.
- Taschereau-Dumouchel, V., Kawato, M., & Lau, H. (2019). Multivoxel pattern analysis reveals dissociations between subjective fear and its physiological correlates. *Molecular psychiatry*, 1-13.